# Fossil evidence for vampire squid inhabiting oxygen-depleted ocean zones since at least the Oligocene

Martin Košťák [1][✉], Ján Schlögl[2], Dirk Fuchs[3], Katarína Holcová[1], Natalia Hudáčková[2], Adam Culka [4], István Fözy[5], Adam Tomašových [6], Rastislav Milovský[7], Juraj Šurka[7] & Martin Mazuch[1]

A marked 120 My gap in the fossil record of vampire squids separates the only extant species (*Vampyroteuthis infernalis*) from its Early Cretaceous, morphologically-similar ancestors. While the extant species possesses unique physiological adaptations to bathyal environments with low oxygen concentrations, Mesozoic vampyromorphs inhabited epicontinental shelves. However, the timing of their retreat towards bathyal and oxygen-depleted habitats is poorly documented. Here, we document a first record of a post-Mesozoic vampire squid from the Oligocene of the Central Paratethys represented by a vampyromorph gladius. We assign *Necroteuthis hungarica* to the family Vampyroteuthidae that links Mesozoic loligosepiids with Recent *Vampyroteuthis*. Micropalaeontological, palaeoecological, and geochemical analyses demonstrate that *Necroteuthis hungarica* inhabited bathyal environments with bottom-water anoxia and high primary productivity in salinity-stratified Central Paratethys basins. Vampire squids were thus adapted to bathyal, oxygen-depleted habitats at least since the Oligocene. We suggest that the Cretaceous and the early Cenozoic OMZs triggered their deep-sea specialization.

[1] Institute of Geology and Palaeontology, Faculty of Science, Charles University, Prague, Czech Republic. [2] Department of Geology and Palaeontology, Faculty of Natural Sciences, Comenius University in Bratislava, Mlynská dolina, Bratislava, Slovakia. [3] SNSB-Bayerische Staatssammlung für Paläontologie und Geologie, München, Germany. [4] Institute of Geochemistry, Mineralogy and Mineral resources, Faculty of Science, Charles University, Prague, Czech Republic. [5] Department of Palaeontology and Geology, Hungarian Natural History Museum, Budapest, Hungary. [6] Earth Science Institute, Slovak Academy of Sciences, Bratislava, Slovakia. [7] Earth Science Institute, Slovak Academy of Sciences, Banská Bystrica, Slovakia. [✉]email: martin.kostak@natur.cuni.cz

Oceanic anoxic events record fundamental changes in the structure and functioning of marine ecosystems. They are determined by global carbon-cycle perturbations, warming episodes, reduced ventilation, increased weathering, the acceleration of organic flux export to the seafloor, and/or the isolation of oceanic basins[1,2]. The biotic responses to hypoxic or anoxic conditions in the geological past varied from regional extinctions during oceanic anoxic events[3–5] up to radiations in the wake of anoxia[6–10] and to adaptations to extreme low-oxygen habitats in oxygen minimum zones (OMZs). Understanding the dynamic of these responses can be informative for predicting abundances and geographic distribution of marine species affected by present-day trends in deoxygenation driven by human activities. The Recent deep-sea vampire squid *Vampyroteuthis infernalis*[11] that inhabits the OMZs in the Atlantic, Indian, and Pacific Oceans possesses extraordinary adaptations to low oxygen concentrations, including a low metabolic rate[12,13] and a detritivorous trophic strategy[14], in contrast to predatory strategies of most other cephalopods. *Vampyroteuthis* is characterized by a mosaic of characters of the superorders Decabrachia and Octobrachia (Octopodiformes or Vampyropoda in other terminologies), but morphological[15] molecular[16–18] and combined studies[19,20] indicate that *Vampyroteuthis* belongs to the octobrachian lineage. However, it is unclear when species of the family Vampyroteuthidae evolved their unique adaptations as no Cenozoic species within the vampyromorph lineage were described until now. This gap indicates not only a major preservation bias (Lazarus effect), but also inhibits any inferences about the timing of deep-sea colonization by vampire squids. The Lazarus effect[21] can either reflect a decline in the outcrop area of post-Cretaceous deep-sea oxygen-depleted habitats and/or a decline in geographic range or in total population size of vampire squids so that their preservation potential is reduced even when the outcrop area remains the same.

The succesive shift of cephalopods into deeper part of oceans tends to be explained with hypotheses that postulate exclusion from shallower habitats driven by higher biotic or abiotic pressures[22,23]. The first hypothesis suggested that coleoids in shallower waters were effectively outcompeted and colonized deeper environments with smaller biotic pressures (explaining survivory strategy in nautiloids). Some lineages were subsequently able to reinvade shallower waters[22]. The second hypothesis[23] suggested that although coleoids inhabited both shallow and deep habitats, extinctions preferentially occurred in shallower environments, and the temporal shift in the preference for deeper habitats is simply indirectly driven by higher extinction rate in shallower environments. Hereby, we suggest that active specialization to deep-sea habitats with anoxic conditions indicate that bathymetric variability in origination rate is also important in explaining the long-term trends in the bathymetric distribution of octobrachians.

Soft part morphologies indicate that Mesozoic gladius-bearing coleoids belong, as the vampire squid, to the Octobrachia[24,25]. The suborder Loligosepiina that diverged during the Triassic from the precursors of the Octopoda[26], represents the vampyromorph branch that led to the origin of vampire squids[27]. The last stratigraphic record of loligosepiids corresponds to the Lower Aptian (Lower Cretaceous). Therefore, the fossil gap between the Recent *Vampyroteuthis* and its Cretaceous ancestors is at least 120 My. However, here we argue that an enigmatic fossil described by Kretzoi[28] as *Necroteuthis hungarica* from the Oligocene Tard Clay Formation of the Hungarian Paleogene Basin (HPB) belongs to this vampyromorph lineage and fills the gap in the fossil record. Kretzoi[28] and subsequent authorities regarded the morphology as a squid gladius[29], whereas other authors thought to have identified a sepiid cuttlebone[30]. This uncertainty is rooted in the fact that fossil gladii appear to be primarily mineralized.

Only chemical analyses (or X-ray diffraction) are able to distinguish between aragonite (the main component of cuttlebones) and francolite (modification of apatite), which is today widely seen as the diagenetic product of an originally chitinous gladius. The chemical composition and thus the real systematic affinities of *Necroteuthis* remained controversial because the single holotype specimen was lost since Kretzoi's first investigations. However, we rediscovered the holotype of *N. hungarica* in the collection of Hungarian Natural History Museum in 2019, and micro-computed tomography (μ-CT) and SEM investigation unambiguously demonstrate that the specimen does not correspond to a cuttlebone, but instead represents a gladius of an Oligocene octobrachian. Owing to a triangular median field flanked by well-developed hyperbolar zones, *Necroteuthis* is most likely affiliated to both extinct loligosepiids and extant *Vampyroteuthis*[25,31]. The completely preserved gladius does not suggest any long-distance transport, and the absence of predation and epibiont activities indicate short residence time in the floating phase or at the sediment–water interface.

Our goals are (1) to examine the gladius of *N. hungarica* using SEM, μ-CT, geochemical analyses, and comparative anatomy, allowing us to properly assign this species, (2) to reconstruct the Oligocene environments inhabited by this cephalopod, and (3) and to track the onshore–offshore shift of vampyromorphs to bathyal and oxygen-depleted conditions since the Early Jurassic to Recent. A geographically extensive oxygen-depleted ecosystem that was established during the initial Early Oligocene isolation of the Parathetys (with salinity stratification, coccolith blooms, and bottom-water anoxia also documented in the Austrian Molasse Basin and in the Outer Carpathians[32,33]) could have triggered the adaptation of this species to deeper portions of basins. This analysis allows us to assess whether the extant *Vampyroteuthis* migrated to the deep-sea associated with dysoxic/anoxic conditions only recently.

## Results

**General gladius morphology**. The almost complete gladius from Hungary (Supplementary Fig. 1), exposed in dorsal view, is 146 mm long and 60 mm wide (Fig. 1). It consists of a triangular median field (with a rounded anterior margin) laterally flanked by a pair of lateral fields, and a pair of hyperbolar zones (Fig. 1d). These characters are diagnostic for the suborder Loligosepiina[26]. The gladius is markedly flattened owing to the compaction. The undeformed gladius of *Vampyroteuthis* encircles the viscera for ~180° (ref. [34]). *Necroteuthis* differs from loligosepiids in having shorter hyperbolar zones. The hyperbolar zone length of *Necroteuthis* is instead similar to *Vampyroteuthis*. *Necroteuthis* and *Vampyroteuthis* moreover share the unique existence of a posterior process that covers and extends the conus. The posterior part of the gladius shows markedly increased thickness, caused by the presence of compacted conus.

**Micro-CT analysis**. The μ-CT analysis (Fig. 2a–i) shows no evidence of a ventral chambered part. Therefore, the structure under investigation represents a gladius and not cuttlebone, in contrast to suggestions of previous authors[30]. The thickness of the gladius increases posteriorly (Fig. 2e–i). The maximum thickness (for ~2 mm) is located in the conus part (Fig. 2i). The ventral view demonstrates again the triangular and fattened character of the median field, the ventrally reduced conus part and the well-distinguished lateral fields.

**SEM analysis**. The gladius is multi-laminated (Fig. 3), typical for both decabrachian and octobrachian gladiuses[25]. Ultrastructurally, the laminae are composed of amorphous matter.

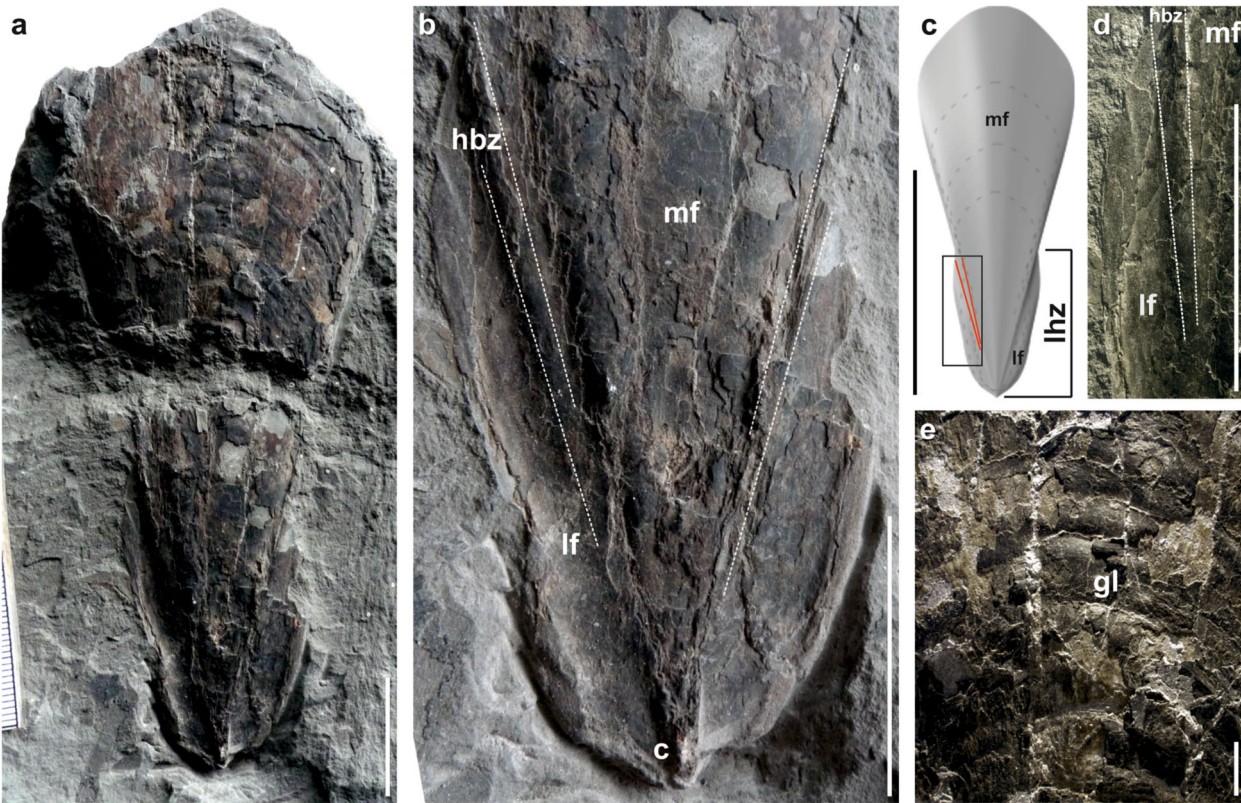

**Fig. 1 Gladius of *N. hungarica* Kretzoi, 1942 (holotype—specimen no. M59/4672 Hungarian Natural History Museum). a** Nearly complete gladius, dorsal view, scale bar = 2 cm. **b** Detail of the apical part forming conus (c), mf median field, lf lateral fields, hbz hyperbolar zones, dashed lines mark the hyperbolar zones separating lateral fields from the median field, scale bar = 2 cm. **c** Reconstruction of the gladius, red lines demarcate hyperbolar zones, mf enlarged median field, lhf length of hyperbolar zones, scale bar = 10 cm, rectangle shows the postion of the **d**, scale bar = 2 cm. **d** Detail of the lateral field with hyperbolar zone and median field, scale bar = 2 cm. **e** Detail of the median field with marked concentric growth lines (gl).

**FTIR**. The chemical composition analysis based on the Fourier transform infrared spectroscopy (FTIR; Supplementary Fig. 2) corroborates the presence of organic material (spectrum E in Supplementary Fig. 2). The bands at 2963, 2930, and 2878 cm$^{-1}$ are attributed to the antisymmetric and symmetric stretching vibrations of -CH$_3$ and -CH$_2$- functional groups. Some more diagnostic bands that were used for chitin identification, such as amide I and amide II bands at 1649 and 1544 cm$^{-1}$ as reported in literature[35], were not properly resolved. Gypsum (Supplementary Fig. 2A, B) and (hydroxyl)apatite (Supplementary Fig. 2C, D) were unambiguously identified in the samples.

**Micropaleontological analysis**. The sediment fraction 0.063–2 mm contains the framboidal pyrite and abundant skeletal remains, mainly fish bones and fragments of fish scales. Fish teeth, the organic-walled acritarchs *Leiosphaeridia*, *Tasmanites*, dinocysts, and small benthic foraminifera (genera *Caucasina*, *Miliammina*, *Fursenkoina*, and *Bulimina*) were present. Shells of foraminifers attain 44% of the microfossil assemblage, whereas organic-walled algae and acritarchs (*Tasmanites* with 80%, *Wetzelielloideae* with 9%, *Leiosphaera*) contribute with 56% (Supplementary Fig. 3). The benthic foraminiferal assemblage consists of nine taxa (Shannon index is 0.74, sample size = 68) and is strongly dominated by *Caucasina oligocaenica* (80.4%) and *Fursenkoina subacuta* (12.4%). The benthic foraminiferal oxygen index (BFOI) reaches −48 and indicates dysoxic conditions, with values of dissolved oxygen between 0.1–0.3 ml/l (ref. [36]).

The calcareous nannoplankton assemblage is dominated by *Reticulofenestra lockeri*, *Coccolithus pelagicus*, and *Reticulofenestra ornata*. The assemblage ranges from extraordinary well-preserved coccoliths (Fig. 4) to coccoliths that are strongly degraded and overgrown by carbonate material. The diversity of nannoplankton assemblages is low to medium, with 17 species recorded in total and 6–11 species per sample, without any reworked taxa (Supplementary Table 1).

**Stable isotope analysis**. Sediment samples used for the stable isotope analysis ($\delta^{13}$C, $\delta^{18}$O) were drilled from a homogeneous, diagenetically unaltered component of sediment. Samples with bioclasts, signs of recrystallization, cements, and carbonate veins were excluded. The bulk carbonate samples show no significant correlation between $\delta^{13}$C and $\delta^{18}$O ($r \sim 0.33$) and are isotopically close to bulk samples from the Tard Clay Formation analyzed by ref. [37]. High resolution sampling of 2.5-mm thick sediment increments from a 4 cm long transect surrounding the gladius showed negative values of $\delta^{13}$C$_{carb}$ (−3.05 to −2.08‰), and highly negative values of $\delta^{18}$O (−5.71 to −6.70‰; Fig. 5). In contrast to *Necroteuthis*-bearing sediment from the Tard Clay Fm., the $\delta^{13}$C$_{carb}$ and $\delta^{18}$O$_{carb}$ from cuttlefish *Archaeosepia*-bearing sediments of the overlying Kiscell Clay Fm. show less negative values ($\delta^{13}$C$_{carb}$ average = −0.78‰ and $\delta^{18}$O$_{carb}$ average = −2.78‰; Supplementary Table 2). The values of $\delta^{13}$C$_{org}$ of the *Necroteuthis*-bearing sediment are relatively uniform, ranging between −27.07 and −26.46 (light gray, lamina sample no. 14, Fig. 5). The $\delta^{13}$C$_{org}$ directly from the sediment yielding gladius (Fig. 5, samples 11–13) reaches −26.73‰ to −26.74‰ (Supplementary Table 3).

## Discussion

The gladius of *Necroteuthis* exhibits a mosaic of gladius features characteristic of Mesozoic loligosepiids and the extant genus

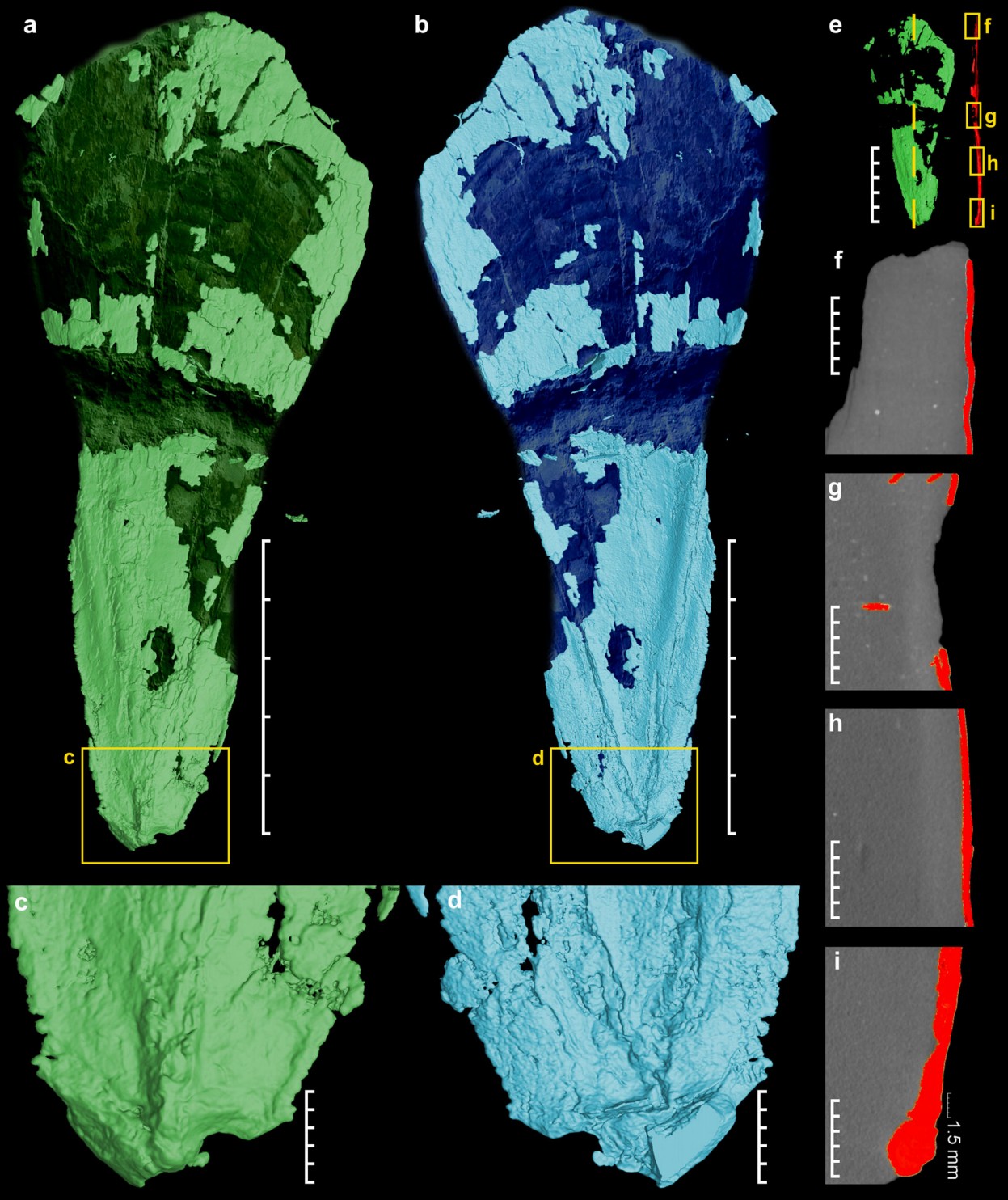

**Fig. 2 Micro-CT visualization of the gladius *N. hungarica*. a** Dorsal view. **b** Ventral view showing typical traingular median field (scale bars **a**, **b** = 5 cm). **c** detail of the posterior part forming conus, dorsal view. **d** detail of the posterior part lateral fields expansion, ventral view (scale bars **c**, **d** = 1 cm). **e** Position of lateral micro-CT sections (**f–i**), scale bar = 5 cm. **f–i** Lateral gladius sections (red color) documenting rise of thickness towards the apex (conical part), scale bars = 0.5 cm.

*Vampyroteuthis*. We can exclude affinities of *Necroteuthis* to teudopseid or prototeuthid octobrachians because the former are characterized by a pointed median field and the latter by conspicously shorter hyperbolar zones[17]. Also, *Necroteuthis* does not show any similarities to the gladii of teuthid decabrachians. Teuthid gladii are more flimsy and markedly more slender than gladii of vampyromorphs.

Both *Necroteuthis* and *Vampyroteuthis* share comparatively short hyperbolar zones that are weakly arcuated[25,31]. The suborder Loligosepiina by contrast is typified by long to very long

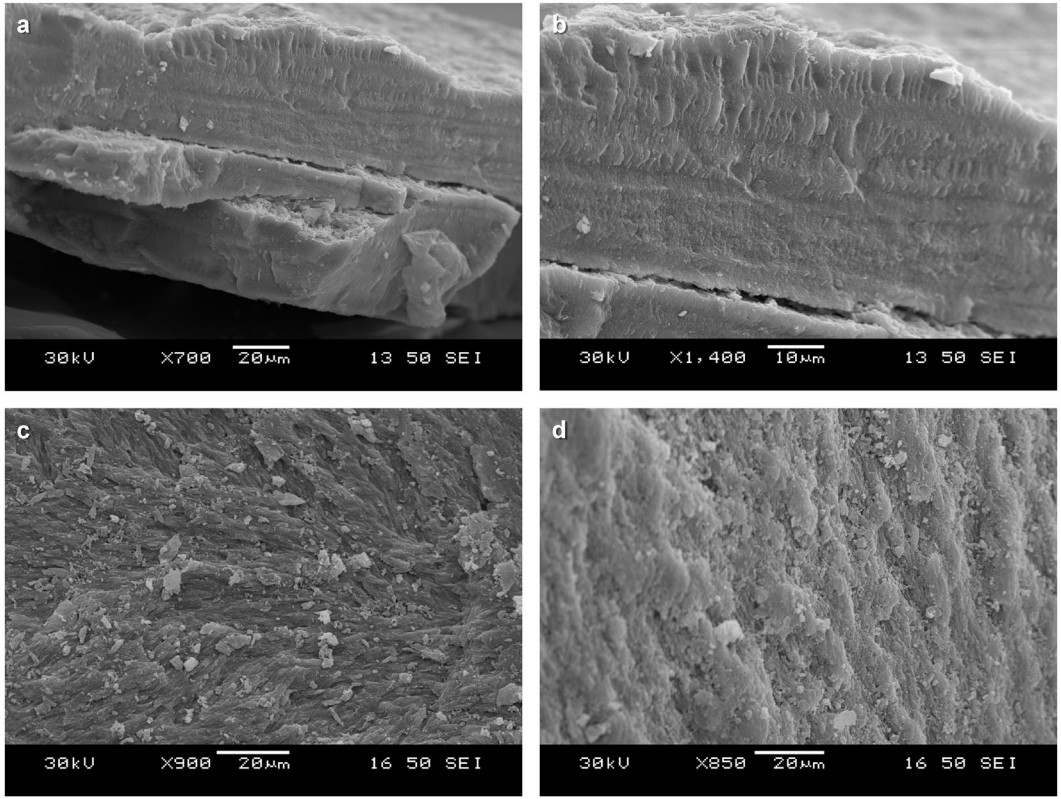

**Fig. 3 SEM photos of the gladius *N. hungarica*. a** Complete thickness of the gladius in the median field area (central part of the gladius) with marked lamination, identical to the Recent gladius-bearing coleoids. **b** Detail of laminated gladius. **c**, **d** Details of the gladius surface. Note, the original β-chitin material has predominantly been replaced by (hydroxyl)apatite and gypsum during diagenetic and postdiagenetic processes.

well arcuated hyperbolar zones. Moreover, many loligosepiids exhibit a concave gladius margin rather than a distinctly convex one. Finally, the shared possession of a posterior process is unique. The latter mutualities suggest closer affinities of *Necroteuthis* to *Vampyroteuthis* than to loligosepiids. The assumed phylogenetic relationships are shown in the Fig. 6.

The broad and weakly specific bands within the FTIR probably reflect a combination of the signal of the organic residue and the newly formed diagenetic minerals. The -CH$_3$ and -CH$_2$- functional groups suggest the presence of organic matter within the gladius, i.e., fragments of original chitin composition. Diagnostic bands (amide I and amide II bands at 1649 and 1544 cm$^{-1}$) are overwhelmed by the signal from diagenetic minerals. We assume that the original chitinous matter was replaced by (hydroxyl) apatite as has also been suggested for mesozoic gladii[38]. This replacement is probably coupled with early diagenetic phosphatization as documented also by the presence of phosphatized coprolites (Fig. 5) in the surrounding sediment. The spectra document a mixture of fossilized organic matter with gypsum and (hydroxyl)apatite (Supplementary Fig. 2E).

The deposition of the Tard Clay Fm. took place in several hundreds of meters-deep HPB during the Early Oligocene[39] as a relatively high rate of subsidence of the HPB coupled with an overall increase in eustatic sea level was not strongly compensated by sediment accumulation. Sedimentation of the Tard Clay Fm. occurred at ~400 m in the western part of the basin during the NP23 Zone (borehole Kiscell-1) and at even larger depths ~800 m in the eastern parts of the basin (borehole Cserepvaralja, based on detailed tectonical and sedimentological analyses)[39]. The water-column stratification with low-salinity and nutrient-rich surface layer and with oxygen depletion on the bottom in the HPB is indicated by benthic–pelagic gradients in δ$^{18}$O, with highly

negative δ$^{18}$O values in planktonic foraminifers[40] and in the bulk coccolith-derived carbonate (as observed here), by blooms of stress-tolerant nannoplankton assemblage that are typical of low salinity[41], by weak levels of bioturbation and by low BFOI values in the whole Tard Clay Fm[42]. Both micropaleontological and geochemical data obtained from the surrounding sediment confirm this anoxic or even euxinic model for the deposition of the Tard Clay Fm., as previously interpreted by refs. [40,43].

First, blooms of calcareous nannoplankton dominated by the endemic Parathethys species *R. ornata* indicate very low-salinity and/or high nutrient levels in the surface layer and high abundance of *R. lockeri* indicates hyposaline waters[44]. Our SEM studies show that the carbonate sediment associated with *Necroteuthis* is partly represented by coccoliths. The extremely negative δ$^{18}$O from bulk rock sample thus probably reflects the composition of surface water, in which coccoliths were calcified. These values are in accordance with δ$^{18}$O of planktonic foraminifera[40] that precipitated calcite in the same surface waters as calcareous nannoplankton. The difference in δ$^{18}$O between co-occurring planktonic and benthic foraminifers[40] suggest that the isotopic values were not modified by diagenesis and that the difference reflects a strong salinity stratification in the basin, with the prominent influx of freshwater and the formation of hyposaline surface waters. The water-column stratification was most likely driven by freshwater inflow, but poor ventilation was probably also induced by incipient isolation of Parathethys in the Early Oligocene[32,33].

Second, the TOC values in the Tard Clay Fm. (~3%)[37] indicate high primary productivity supported by blooms of algae (*Tasmanites*, *Leiosphaera*, and taxa of Wetzelielloideae) coupled with inefficient recycling of organic matter on the seafloor. High abundance of dinoflagellates of the genus *Wetzeliella*, high values

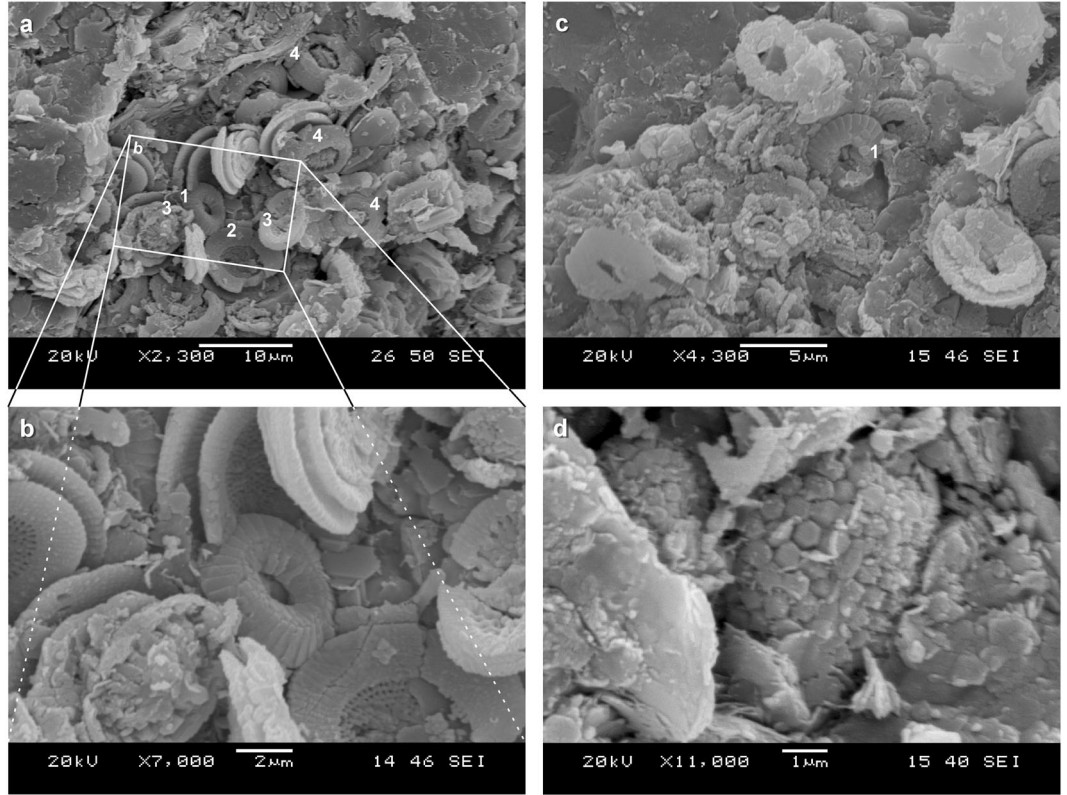

**Fig. 4 SEM photos of calcareous nannofossils at *Necroteuthis* gladius level. a–c** Partly exposed nannoplankton accumulation with well-preserved coccoliths. **a**—general view. **b** Detail: **a**—*C. pelagicus*, **b**—*R. umbilicus* (probably part of coccosphaera), **c**—*Pontosphaera enormis*, and **d**—*R. ornata*. **c** Corroded and recrystallized undeterminable calcareous nannoplankton with exception of dissolution-resistant *C. pelagicus* (**a**). **d** Framboidal pyrite as indicator of bacterial activity in dysoxic sediment.

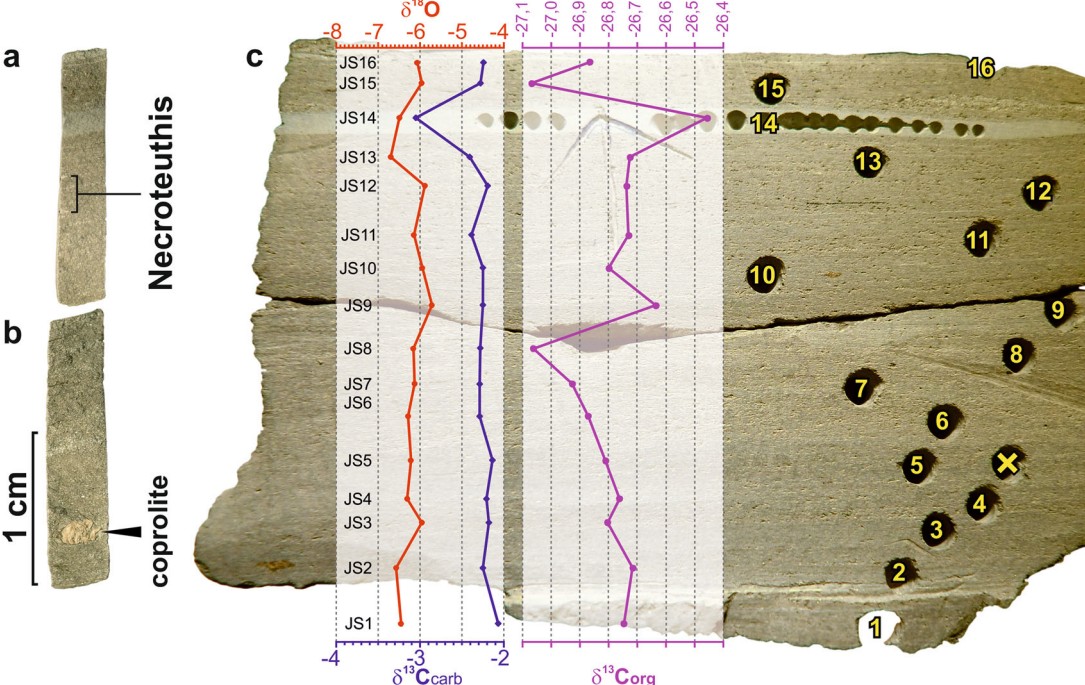

**Fig. 5 Transect trough the gladius-bearing sediment (scale bar = 1 cm) in relation to stable isotope data of the $\delta^{18}O_{carb}$, $\delta^{13}C_{carb}$, and $\delta^{13}C_{org}$. a** Upper part with position of the gladius level (equals to JS12, probe no. 12). **b** Lower part with a phosphatized coprolite. **c** Positions of probes used for geochemistry. Sample no. 14 (JS14) with markedly different both $\delta^{13}C$ isotope values is represented by light-gray mudstone lamina.

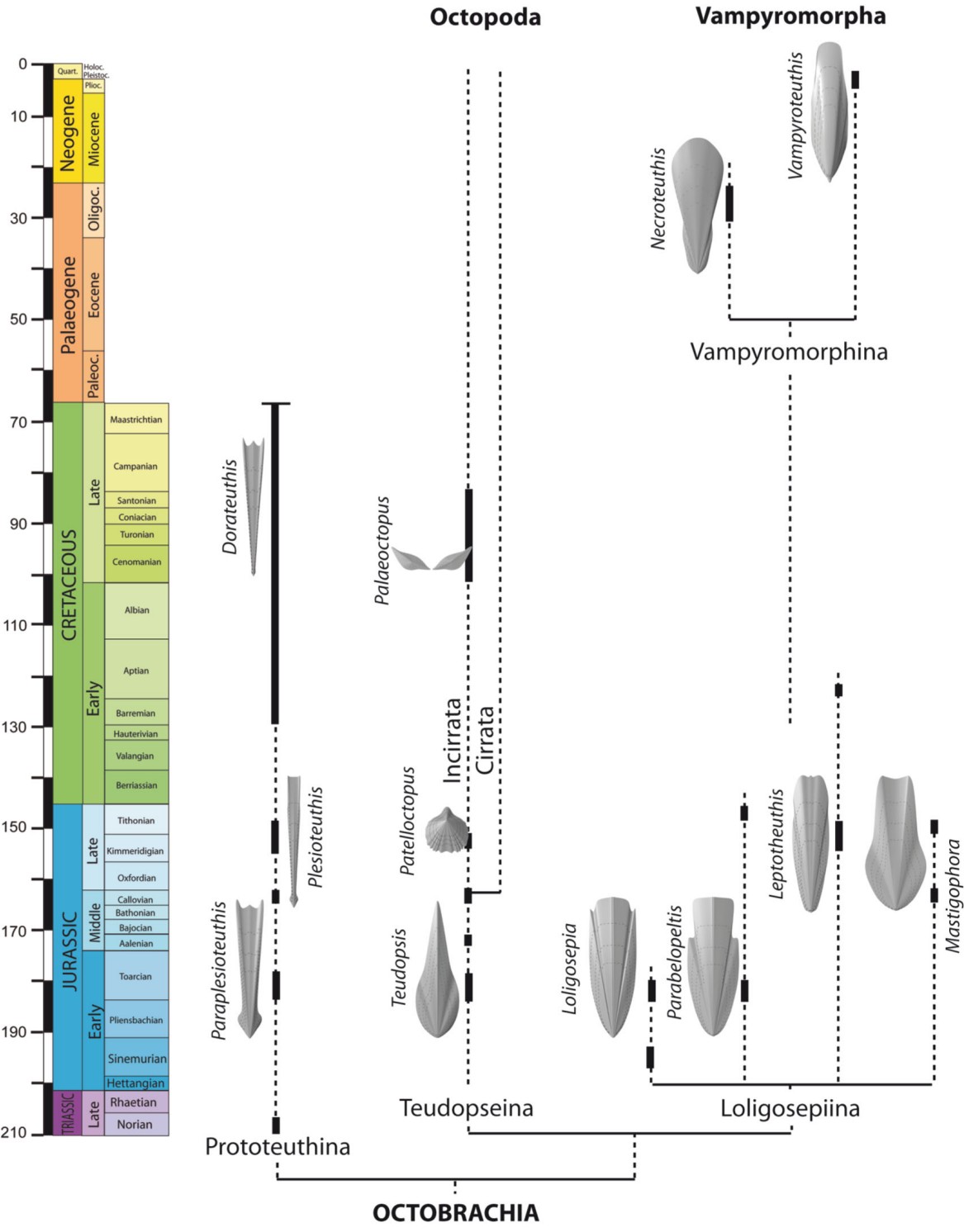

**Fig. 6 Phylogeny and stratigraphic distribution of the Octobrachia.** The presumed relationship between loligosepiid suborders Vampyromorphina and Loligosepiina is indicated. Note, not all genera within Loligosepiina are figured (after[26], this paper).

of TOC, the lack of bioturbation, and negative BFOI values in the sediment sample with the vampire squid further corroborate that *Necroteuthis* was deposited during a period characterized by eutrophication in the surface layers and by anoxic conditions on the seafloor. Although previous authors[40] found that the BFOI varying between −40 and 0 indicates suboxic conditions (with ~0.3–1.5 ml/l dissolved oxygen content in the bottom waters), our assemblage is strongly dominated by triserial foraminiferal forms

(*Caucasina*, *Bulimina*, and *Fursenkoina*) belonging to deep infaunal species tolerating hypoxic conditions[36,45], and BFOI is thus highly negative and attains −48. Such values indicate dissolved oxygen between 0.1 and 0.3 ml/l (ref. [36]). Hypoxic conditions are also documented by very thin, densely porous shells of *Caucasina* and *Fursenkoina* (5–8 pores/μm²). Accumulations of organic-walled algae of the genus *Tasmanites* were reported from dysoxic environments[46].

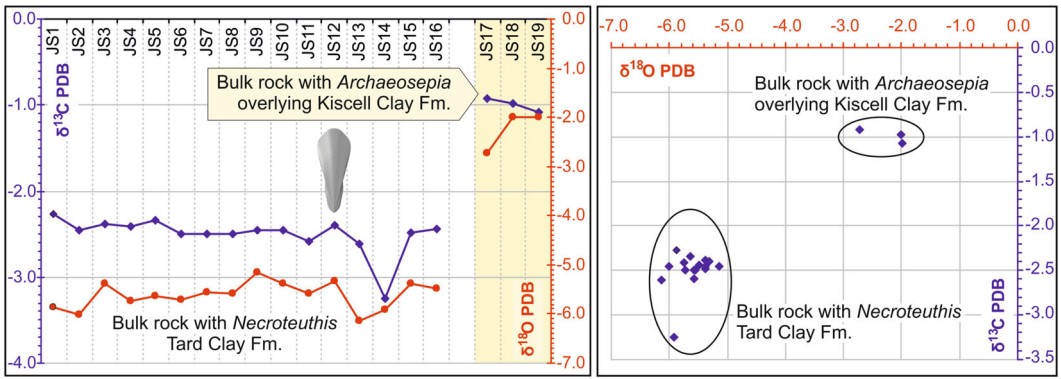

**Fig. 7 Differences in the $\delta^{13}C_{carb}$ and $\delta^{18}O_{carb}$ isotopic signal from loligosepiid *Necroteuthis*-bearing sediment of the Tard Clay Fm.** Samples (JS1–JS16) and overlying, sepiid *Archaeosepia*-bearing sediment from the Kiscell Clay Fm. (samples JS17–JS19). More positive values of $\delta^{13}C_{carb}$ and $\delta^{18}O_{carb}$ represent more oxygened normal marine environment in younger strata. An accurate level of of *Necroteuthis* gladius is indicated by picture (JS12).

As the Recent vampire squid habitat is exclusively stenohaline (as other coleoid cephalopods), *Necroteuthis* very likely did not migrate to the upper surface layer with very low salinity. The stable isotope record from bulk rock samples from the overlying Kiscell Clay (spanning Rupelian/Chattian boundary) with mineralized cuttlebones of the sepiid *Archaeosepia* demonstrate that this cuttlefish lived in shallower and more oxygenated environments, as evidenced by abundant molluscs, decapod crustaceans, ostracods, brachiopods, and echinoderms[47] and by the $\delta^{18}O_{carb}$ isotopic signal that indicates a weaker water-column stratification (Fig. 7). Therefore, *Necroteuthis* inhabited oxygen-depleted environments, whereas latter sepiids inhabited well-oxygenated environments. The expansion of other sepiids to deeper habitats was probably functionally constrained because the deepest records of few extant sepiid taxa reach 400–500 m (ref. [48]) and functional analyses indicate that a gas-filled cuttlebone is subject to shell implosion at larger depths. The majority of sepiids therefore prefer waters shallower than 150 m and avoid greater depths[48].

The offshore or bathyal retreat documented in the distribution of many lineages of marine invertebrates during the Mesozoic and Cenozoic is explained either by competitive and predatory exclusion from onshore habitats[49], and/or by a push toward offshore as organic flux to deeper habitats increased during the Cretaceous and the Cenozoic[50]. On the one hand, although the roots of some present-day deep-sea invertebrate lineages, such as ophiomycetid ophiuroids or pterasterid and benthopectinid asteroids, can be traced back to the Jurassic[51], the role of ocean anoxic events and the recurrent expansion of OMZs during oceanic anoxic events is mainly used to invoke repeated extinction risk disproportionately affecting deep-sea faunas adapted to normoxic conditions[52]. On the other hand, the OMZ can also trigger range expansions, speciations driven by opportunities with smaller predation pressure[53], and can lead to endemism at macroevolutionary time scales[54,55] and to habitat specialization allowed by persistent hypoxic conditions[56] and/or by steep spatial gradients in oxygen concentrations[57]. However, morphological novelties based on cryptic specializations may be generated under anoxic conditions in both nearshore and offshore habitats[58]. Our compilation below documents (1) that the bathyal shift of vampyromorphs took place at least since the Oligocene or earlier during the Cretaceous and (2) that it might coincide with the development of the OMZ in the Central Paratethys. We thus invoke the role of OMZ as a trigger for habitat specialization of vampyromorphs. It is notable that the majority of Jurassic and Cretaceous loligosepiid records are associated with hypoxic or anoxic conditions (Fig. 8) that characterized poorly ventilated epicontinental seas of the NW European shelf (see below).

We propose the following geological timeline by documenting the occurrences of vampyromorphs in multiple Mesozoic Lagerstätten to explore the potential factors that may have driven vampyromorphs to adapt toward oxygen-depleted conditions in deep-sea environments.

**Toarcian oceanic anoxic event**. Jurassic loligosepiids show a relatively high diversity and abundance in black shales with exceptional preservation (i.e., Lagerstätten in southern Germany, Luxemburg, France, and UK) that were deposited on the NW European epicontinental shelves during the Toarcian Ocean Anoxic Event. Anoxic conditions were associated with warming, reduced ventilation, increased weathering on land, and increasing of freshwater influence, with the development of haline stratification in some basins[59]. It is difficult to assess whether the Early Jurassic loligosepiids (*Loligosepia*, *Jeletzkyteuthis*, *Geopeltis*, and *Parabelopeltis*) either lived within these anoxic basins (characterized by repeated but short-term reoxygenation events) or inhabited shallower, better-mixed normoxic environments. However, semi-enclosed oceanic basins in the Tethys and the Panthalassa were also characterized by anoxia and by the deposition of organic-rich sediments, but loligosepiids were not yet recorded from these deeper, bathyal environments. Therefore, loligosepiids were probably still limited to continental shelves during the Toarcian.

**Callovian La Voulte-sur-Rhône**. The Lower Callovian (Middle Jurassic) locality La Voulte-sur-Rhône yields unusual faunal assemblages. Coleoid cephalopods, including loligosepiids (*Mastigophora*, *Vampyronassa*, and *Proteroctopus*[60]) and prototeuthids (*Romaniteuthis* and *Rhomboteuthis*) represent 10% of macro-invertebrate assemblage preserved in organic-rich marls otherwise dominated by arthropods and echinoderms. The bottom-water conditions at the site of the soft-bottom deposition, with soft-tissue mineralization in the sediment zone with sulfate reduction[61], were probably oxygen-depleted and temporarily anoxic, with limited bioturbation and mass mortalities documented by pavements of epibenthic bivalves (*Bositra*)[62]. As shown in ref. [63], photophilic encrusters are missing, invertebrates are encrusted by non-photozoans groups, such as serpulids, cyrtocrinid crinoids, sponges, and thecideid brachiopods, and the actualistic distribution of sea spiders, some crustaceans and sea stars indicate that the sedimentation took place on the outer shelf at ~200 m with dysphotic or aphotic conditions[64]. However, the co-occurrence of crustaceans with eyes adapted to photic conditions[65] with groups with dysphotic or aphotic preferences indicate

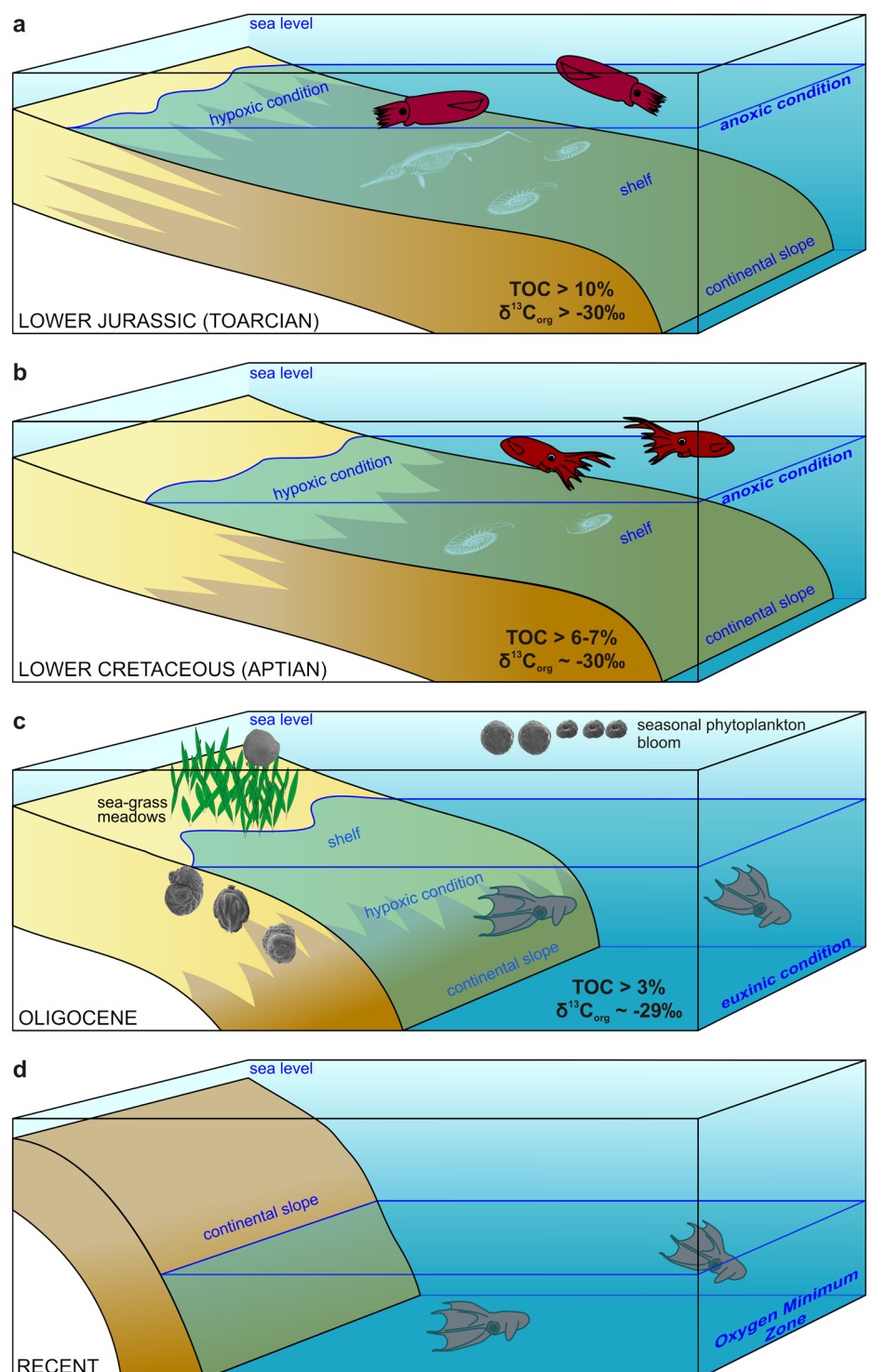

**Fig. 8 Onshore–offshore shift of loligosepiid coleoids in relation to bathymetry and the oxygen minimum zone shift from shelf to deep-sea. a** Toarcian OAE event with loligosepiid occurrences associated with geographically extensive anoxic conditions on shelves. Geochemical data from ref. [59]. **b** Aptian OAE1 event with loligosepiids associated with anoxic conditions on shelves. Geochemical data after refs. [66,67]. **c** Oligocene habitat of *N. hungarica* in the Central Paratethys. Bathymetric conditions of the gladius record correspond to bathyal habitats with depth >400 m (ref. [39]). Shallower, shelf environments were characterized by seagrass meadows with seagrass-associated foraminiferal assemblages. Geochemical data—this study and after ref. [37]. **d** Recent living conditions of *V. infernalis* adapted to the OMZ at depth >600 m (ref. [14]).

that the total assemblage represents a mixture of bathymetrically distinct habitats[66], although any postmortem transport had to be minor and rapid as indicated by the excellent preservation of complete skeletons of fragile organisms. Fault-controlled escarpments with sponge communities not far from the site with the

exceptional preservation[64] indicate steep topographic gradients over short distances, and it is likely that even a limited short-distance migration to soft-bottom habitats in the wake of anoxic events or postmortem transport can explain the mixture of groups differing in ecological requirements in deeper environments. To

summarize, coleoids in these environments still inhabited outer shelf close to the shelf/slope margin during the Callovian, and did not yet expand to deeper bathyal environments.

**Kimmeridgian–Tithonian lithographic limestones**. The Upper Jurassic records from lithographic limestones were deposited in semi-enclosed lagoons characterized by hypersalinity and partly dysoxic regime induced by bacterial activity at the bottom[67]. Although the cephalopod assemblages are predominantly allochtonous in these sediments, they clearly inhabited the surrounding epicontinental seas, and thus also do not record deep-sea conditions. Upper Jurassic loligosepiids are represented by genera *Leptotheuthis*, *Doryanthes*, and *Bavaripeltis*.

**Aptian OAE1**. The Mesozoic record of the Loligosepiina terminates in the Lower Aptian Oceanic Anoxic Event 1a (OAE1) at Heligoland[68]. The laminated, anoxic, and organic-rich "Fisch-schiefer" ("Töck") originated under humid-warm regime with low salinity in upper parts of the water column[69]. The water depth of this habitat has been estimated between 50 and 150 m on the basis of belemnites[69]. The TOC values exceed >6–7% (refs. [69,70]), extreme values to 10% (ref. [71]), the lowest negative $\delta^{13}C_{carb}$ values exceed −9‰ (maximum peak), the lowest $\delta^{13}C_{org}$ ~−30‰ (ref. [69]), reflecting anoxic conditions and salinity driven stratification of the water column. The natural habitat of loligosepiids (*Donovaniteuthis stuehmeri*) is closely tied to oxygen-depleted conditions at the bottom, because low-salinity conditions in the upper parts of the water column are unfavorable for stenohaline fauna. There are no loligosepiid records from similar conditions represented by later OAE2 and OAE3 (Upper Cretaceous). This absence may indicate migration of vampyromorphs into the deep sea at the end of the Early Cretaceous. After this record (OAE1), vampyromorphs disappeared from the fossil record. Our new record from the Oligocene represents the first Cenozoic record of vampyromorphs.

**Oligocene of the Central Paratethys**. While the Mesozoic record of loligosepiids is limited to epicontinental shelves, the Oligocene record of *Necroteuthis* from the Central Paratethys is linked to bathyal environments reaching >400 m and bottom-water hypoxic/anoxic to euxinic conditions[40]. This occurrence thus demonstrates that the shift into bathyal oxygen-limited habitats occurred at least during the Oligocene. The deeper-water bathypelagic conditions probably provided opportunities for expansions of bathymetric ranges. Less predation and competitive pressure is typical of present-day oxygen-limited habitats with high abundance of food supply (especially on the edges of OMZs), where hypoxia-sensitive species are excluded[55]. However, the bathymetric shift of vampyromorph habitats can also be an important feature in the survivor strategy as it may have happened prior to the end-Cretaceous mass extinction. In contrast to ammonite lineages inhabiting continental shelves that were negatively affected by the end-Cretaceous mass extinction, this extinction did not affect deep-sea habitats. For example, some modern cephalopods whose ancestors survived this mass extinction event share a larger size of the eggs (expressed by larger protoconchs) relative to the victims of the mass extinction (with planktotrophic larvae), suggesting that the shift of vampyromorphs to larger water depths, associated with a shift towards production of larger eggs less dependent on planktonic food webs[72–74], occurred prior to the K/T boundary.

**Recent**. The Recent *Vampyroteuthis* inhabits worldwide meso- and bathypelagic zones from 600 to ~1500 m (3300 m at maximum), with the highest abundance between 1260 and 1500 m

(ref. [75]). Its distribution is directly linked to the development of OMZ, which played a crucial role in *Vampyroteuthis* surviving as a refugium until recent times[14].

As we have shown above, the Jurassic and Lower Cretaceous records are associated with hypoxic or anoxic conditions, indicating that vampyromorphs inhabited habitats close to anoxic/euxinic conditions already during the Mesozoic (Fig. 7). Therefore, if vampyromorphs shifted to deeper habitats prior to the K–Pg boundary, their adaptation to hypoxic conditions probably existed already in the Mesozoic, allowing survivorship of the loligosepiid lineage in deep-water refugia linked to the OMZ. The observation that loligosepiids do not occur at other Cretaceous Lagerstätten (e.g., Lebanon and Mexico) that are not directly associated with anoxic events is consistent with this hypothesis. The preservational potential of coleoid bodies in the fossil record is strongly limited by their fragile remains and by large amount of ammonia concentrated in their soft tissues, inhibiting precipitation of authigenic minerals[76]. However, coleoid gladii that represent a taphonomic control for loligosepiids occur in the Lower Turonian shelf sediments (Bohemian Cretaceous Basin) deposited under well-oxygenated conditions[77]. The absence of loliginid gladii suggests that loligosepiids may retreated from shallower environments already during the Cretaceous. Shallow-water sediments representing later multiple ocean anoxic events (OAE2 and OAE3) did not provide any loligosepiids records yet. Deep-water sediments of these periods are considerably rarer, therefore, it is possible that the extent of the preserved deep-sea sediments with exceptional preservation is still insufficient to detect this group[78]. Although it is possible that Mesozoic vampyromorphs were already adapted to low oxygen concentrations, the bathyal habitats of Oligocene *Necroteuthis* stand in sharp contrast to the shelf habitats of Jurassic and Cretaceous vampyromorphs. *Necroteuthis* inhabited bathyal habitats with high primary productivity and bottom-water anoxic conditions[40,43]. When the OMZs in the Central Paratethys basins formed during the Early Oligocene for the first time, the stratified and oxygen-depleted basins potentially generated new opportunities for the vampyromorph range expansion under stable low-oxygen levels.

## Conclusions

The Oligocene *Necroteuthis* was a close relative of the extant deep-sea vampire squid *Vampyroteuthis*. *Necroteuthis* inhabited bathyal depths in the HPB characterized by stratified water-column and oxygen depletion on the bottom. This new insight shows that the Vampyromorpha shifted from shallow to deep water at least during the Oligocene. Adaptations to low concentrations of dissolved oxygen were probably already developed in Mesozoic loligosepiid vampyromorphs, as they inhabited epicontinental shelves with hypoxic conditions; however, the bathyal habitats of Oligocene *Necroteuthis* (~30 Mya) differs markedly from shelf habitats of Mesozoic vampyromorphs below the pycnocline, where competition and predation is typically reduced as hypoxia-sensitive species are excluded, but food supply tends to be high[55].

The coleoid order Vampyromorpha survived major Meso- and Cenozoic oceanic crises, including oceanic anoxies, climatical, and sea-level changes, as well as the major extinction event at the K–Pg boundary. Owing to the scarce fossil record, we are currently unable to estimate the time of divergency between deep-sea Vampyromorphina and shallow-water Loligosepiina suborders. The Vampyromorphina thus consists of two genera, including the Recent genus *Vampyroteuthis* and the Oligocene genus *Necroteuthis*.

The vampyromorph evolutionary strategy reveals competitive advantage and survivory success during global biotic crises, as these niches were not affected by marked environmental changes.

## Methods

**Geological and historical settings**. The gladius of *Necroteuthis* (No. M59/4672 Hungarian Natural History Museum) has been found in a clay pit near Budapest ("Ziegelfabrik von Csillaghegy, NNW—Budapest, Kisceller Ton"). The locality, i.e., the Csillaghegy Brickyard is also known as Péterhegy (or Péterhegy). The clay pit, was refilled and does not exist any more. Two litostratigraphical units were exposed in the clay pit[79] (see Supplementary Information): (1) dark laminated Tard Clay Fm. deposited in a stagnant, restricted basin under anoxic conditions. The Tard Clay is not bioturbated and lacks benthic fauna. (2) The overlying dark gray, bioturbated Kiscell Clay Fm. was deposited under normal marine conditions with water depths between 200 and 1000 m estimated on the basis of benthic foraminifers[80].

In the Csillaghegy clay pit, the two formations are exposed side by side, due to a tectonic contact (fault). Therefore, we assessed in detail whether the gladius comes from the Tard Clay Fm. or the Kiscell Clay Fm. Kretzoi[28] assigned it to the Kiscell Clay Fm., but at that time the Tard Clay Fm. was not distinguished from the Kiscell Clay Fm. However, Kretzoi assigned the age of the Tard Clay Fm. ("Lattorfium oder unteres Rupelium"). Two new lines of evidence confirm that these specimens are derived from this formation. First, the matrix around the *Necroteuthis* specimen is represented by a laminated, hard, argilliferous rock, which is typical of the Tard Clay Fm. Second, co-occurrence of calcareous nannoplankton species *R. ornate*, *Reticulofenestra umbilicus*, and *Discoaster nodifer* indicates the NP22 Zone. This correlation is supported also by absence of *Reticulofenestra abisecta* that appears (first occurrence, FO) in the upper part of the NP23 zone. The repeated blooms of endemic Paratethys algae *R. ornata* took place during the zone interval NP22–NP24 (refs. [44,81]). Therefore, the lithological features and calcareous nannoplankton clearly confirms that *Necroteuthis* specimen originates from the Tard Clay Fm. (NP22 to lower part of NP23 zones).

During the Early Oligocene, the system of the Middle European Paleogene basins was restricted what caused the closure of the seaways toward the weastern Tethys. This new paleogeographic situation was trigerred by orogeny in the South Alpine–Dinaridic belt, in combination with a third or second-order eustatic sea-level drop between 30 and 32 Ma (ref. [39]). These changes led to the origin of the Central Paratethys, initially characterized by widespread deposition of organic-rich shales in stratified basins[82] and by endemism of molluscs[83]. The Tard Clay Fm. was deposited during the Kiscellian under anoxic conditions in the HPB. The palaeobathymetric conditions were analyzed using geobasinal dynamic changes[39] documenting palaeodepth reaching >400 m (i.e., bathyal or deep-sea in biological terminology[78]).

**Micro-CT**. µ-CT imaging was performed with phoenix v|tome|x L 240 device, developed by GE Sensing & Inspection Technologies. Investigated samples were analyzed by using 240 kV/320 W microfocus tube. Scanning parameters were set as follows. For *Necroteuthis* sample: voltage 200 kV, current 250 µA, projections 2500, average 3, skip 1, timing 500 ms, voxel size 80 µm, and 0.5 mm Cu filter. After the scanning process, 3D data sets were evaluated with VG Studio Max 2.2. For the address of storage space, see Supplementary Information (Supplementary Micro-CT imaging).

**Fourier transform Infrared spectroscopy**. The infrared spectra were recorded by micro-ATR technique on a Thermo Nicolet iN10 FTIR microscope, using Ge crystal in the 675–4000 cm$^{-1}$ region (2 cm$^{-1}$ resolution, Norton−Beer strong apodization, MCT/A detector). Standard ATR correction (Thermo Nicolet Omnic 9.2 software) was applied to the recorded spectra. Several miniature samples (<1 mm) of dark-colored fossilized material were analyzed by infrared spectroscopy using a micro-ATR technique. Reference spectra for gypsum and (hydroxyl)apatite were taken from the RRUFF online database of spectroscopic and chemical data of minerals[84].

**Stable isotope record**. Stable C and O isotopes in carbonate fraction of clay sediments were analyzed on isotope-ratio mass spectrometer (IRMS) MAT253, coupled with Kiel IV device for semiautomated carbonate preparation (Thermo-Scientific). A total of 40–100 micrograms of milled powder were loaded into borosilicate glass vials, evacuated, and digested in anhydrous phosphoric acid at 70 °C following method[85]. Yielded $CO_2$ gas was cryogenically purified and introduced into the IRMS via dual-inlet interface. Isotope composition was measured against $CO_2$ reference gas and raw values were calibrated using international reference material NBS18 and two working standards with $\delta^{13}C$ = 5.014‰, +2.48‰, −9.30‰ and $\delta^{18}O$ = −23.2‰, −2.40‰, −15.30‰, respectively. The values are reported as permil vs. PDB, precision of measurement is 0.02‰ for carbon and 0.04‰ for oxygen.

Stable carbon isotopes of organic matter were measured on mass spectrometer MAT253, coupled to elementar analyzer Flash2000 HT Plus (ThermoScientific). Residues after digestion in hydrochloric acid of ~900–1300 micrograms were wrapped into tin capsules and combusted in a stream of helium at 1000 °C in quartz tube packed with tungsten oxide and electrolytic copper. Purified $CO_2$ gas was separated from other gases on capillary GC column (Poraplot Q, Agilent) and introduced into IRMS in continuous flow mode. Raw isotope ratios measured against $CO_2$ reference gas were calibrated to PDB scale, using two international

reference materials (USGS24 carbon, USGS41 glutamic acid) and two working standards, with $\delta^{13}C$ values −16.05, +37.76, −39.79, and −25.60‰, respectively. All the values are reported in permil PDB, precision measured on standards is 0.11 permil. Standard deviation = 0.106‰.

**SEM and imaging**. The fossilized gladius was examined at the Institute of Geology and Palaeontology, Faculty of Science, Charles University in Prague by scanning electron microscope (SEM) JEOL-6380LV at 20, 25, and 30 kV and at ×1.7–10 k magnification. Microscopic gladius remains as well as the bulk rock fragment with Ca-nannofossils were coated with gold and investigated in the low and high vacuum modes. The macro-photo of *Necroteuthis* specimen was taken using the camera Canon EOS 600D. Photographs were improved using CorelDRAW X7 and Corel Photo-Paint X7 graphic softwares.

**Microfossil investigation**. Microfossils were collected and determined using Olympus SZ61 binocular stereoscopic microscope Olympus B750 and optical Zeiss Axiolab 5 microscope, and documented by SEM microscope JEOL-6380LV. Determination of foraminifers is in accordance with published methodics[86]. Paleoecological parameters were evaluated on the presence and dominance of taxa exhibiting special environmental significance[36,46]. The BFOI[36] was determined to assess bottom-water concentrations of dissolved oxygen. The index is based on proportion of oxic, suboxic, and dysoxic indicator species of benthic foraminifera. The Kaiho's classification[36] of benthic foraminifera to these three groups was used. Because at least one oxic species was found, we used the following equation for counting of the BFOI in agrement with published data[36]:

$$\{[O/(O + D)]' \, 100\}$$

Where: *O*—proportion of oxic species and *D*—proportion of dysoxic species. Values of the BFOI index can vary from −100 to +100, higher the values indicate higher oxygen concentration.

Taphonomic analysis of foraminiferal assemblages was performed following the concept of refs. [87,88]. Beside the foraminifers, the organic-walled cysts and algae were counted in wash residuum. Calcareous nannoplankton was studied by both optic microscope (magnification 1000×, crossed and parallel nicols) and SEM. Slides for optic microscopy were prepared according to ref. [89].

**Reporting summary**. Further information on research design is available in the Nature Research Reporting Summary linked to this article.

## Data availability

The micro-computed tomography images generated during and/or analyzed during the current study are available in the figshare repository: https://doi.org/10.6084/m9.figshare.13526024 (ref. [90]). The *Necroteuthis* holotype which is the subject of the imaging is housed in the Hungarian Natural History Museum under item: No. M59/4672.

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

## Acknowledgements

This work is supported by the projects: PROGRES Q45 (M.K., K.H., M.M., and A.C.), Center for Geosphere Dynamics (UNCE/SCI/006—M.K., K.H., M.M., and A.C.), 20-05872 S (K.H.), 18-05935 S (K.H.), GAČR No. 20-10035 S (M.K. and M.M.), APVV 17-0555 (J.S., N.H., and A.T.), and VEGA 0169-19 (J.S., N.H., and A.T.).

## Author contributions

M.K. and J.S. had initial idea to describe the material, investigation, conceptualization, and data interpretations. M.K., J.S., D.F., I.F., K.H., N.H., A.T., A.C., R.M., and J.Š. performed investigation, methodology, writing—original draft, writing—review and editing, with input of all authors. K.H. and N.H.: micropalaeontological analysis. A.C., R.M., and J.Š.: geochemical analysis. M.M.: visualization and SEM analysis. A.T.: proofread and corrected the entire text, data interpretations.

## Competing interests

The authors declare no competing interests.
