## [Peer Review File · Communications Biology]

Reviewers' comments:

Reviewer #1 (Remarks to the Author):

This is an innovative contribution using a multi-proxy approach to understand the establishment of modern deep-sea communities and therefore not only of great interest to paleobiology but biology more generally. The authors provide the first robust evidence that vampire squids colonized the deep sea only rather recently. There are just some minor points I would like to see addressed before publication:

1) Establishment of deep-sea coleoids: traditionally, two hypotheses could explain the preferential occurrence of vampyromorphs in the deep sea (or coleoids versus nautiloids more generally) which are not mutually exclusive (Nixon et al., 2003; Packard, 1972) – shift in distribution from shallow water and/or occurrence in both deep and shallow water with extinction in shallow water. It is quite interesting that what you are suggesting here is also to some degree reminiscent what happen with the nautiloid lineage which was initially more restricted to shallow water. It is hard to distinguish between these scenarios in the fossil record (Oji, 2001) but you provide quite good arguments for the former. I would like to see these two hypotheses/possibilities a bit more development in the introduction with historical references. In this context, also the latest research concerning the importance of shifts in redox conditions you are already alluding too could also be integrated (Wood and Erwin, 2018).

2) Preservation and collection potential of vampyromorphs: I would be crucial to have a little bit more background on the preservation and collection potential of vampyromorphs versus Decabrachia and for coleoids more generally (Clements et al., 2017). A bit more reference that historically various shells have been wrongly attributed to Decabrachia would also be helpful to place your study into context. This is trivial for coleoid workers but would make even more relevant for researchers beyond this field.

3) Please provide essential data from the tomography: The tomography of the specimen is crucial to reject its previous assignment to Sepiidae and to be able to scientifically reproduce your results. It would be standard practice to at least provide the raw full resolution image stack and metadata and recommended to also provide the prepared dataset (Davies et al., 2017). Platforms like Zenodo and other more specialized databases (e.g., morphosource.org) provide storage space and wide range of possibilities.

4) Additional information on the degree of diagenesis and preservation (e.g., cements) of isotope samples: it would be crucial to have some additional background information on the preservation and diagenesis (Cathodoluminescence) of the samples using for isotope analysis.

5) Additional references: some additional statements are made (concerning indicators of little transport of the fossil) and definition used (e.g., Lazarus effect) which needs to be backed up with references.

These and additional suggestions can be found in the annotated pdf.

Suggested references:

Clements, T., Colleary, C., De Baets, K., and Vinther, J., 2017, Buoyancy mechanisms limit preservation of coleoid cephalopod soft tissues in Mesozoic Lagerstätten: *Palaeontology*, v. 60, no. 1, p. 1-14.

Davies, T. G., Rahman, I. A., Lautenschlager, S., Cunningham, J. A., Asher, R. J., Barrett, P. M., Bates, K. T., Bengtson, S., Benson, R. B., and Boyer, D. M., 2017, Open data and digital morphology: *Proceedings of the Royal Society B: Biological Sciences*, v. 284, no. 1852, p. 20170194.

Nixon, M., Young, J. Z., and Young, J. Z., 2003, *The brains and lives of cephalopods*, Oxford University Press.

Oji, T., 2001, Deep-Sea Communities: *Palaeobiology II*, p. 444-447.

Packard, A., 1972, Cephalopods and fish: the limits of convergence: *Biological Reviews*, v. 47, no.

2, p. 241-307.

Wood, R., and Erwin, D. H., 2018, Innovation not recovery: dynamic redox promotes metazoan radiations: *Biological Reviews*, v. 93, no. 2, p. 863-873.

Reviewer #2 (Remarks to the Author):

Really really like this paper! It answers an important question for cephalopod researchers and palaeontologists – how and why is *Vampyroteuthis* a single taxon, highly specialised octopod and how does it fit evolutionarily into the evolution of cirrate octopus? I think this paper discusses an important fossil and proposes some really exciting novel hypotheses about the evolution of these fantastic and interesting animals. I strongly endorse this paper and think it could get significant press interest because it studies such charismatic and popular animals. The paper is well written and uses appropriate techniques. I look forward to citing this paper in the future!

Thomas Clements

I have attached a word doc and a PDF with my comments.

Reviewer #3 (Remarks to the Author):

The article presents a new description of a cephalopod fossil, *Necroteuthis* from the Oligocene of Hungary, which had previously been attributed to a cuttlefish or a squid gladius. This supposedly lost specimen has been rediscovered by the authors and is here reconsidered using new approaches compared to the original description (XR Tomography, SEM, XRD). Since the first description, much work has been carried out on coleoid fossils allowing more comparisons and discussion than in 1942 (numerous articles by Dirk Fuchs, including Chapter 9B of *Treatise on Paleontology* online 2016). The authors attribute the fossil specimen to the *Loligosepiina* and to the family *Vampyroteuthidae*, thus discovering to date, the closest fossil relative to the extant *Vampyroteuthis*.

Following this attribution, the authors study the depositional environment of *Necroteuthis* and conclude to a relatively deep hypoxic habitat. The authors then show that most Mesozoic *Loligosepiids* lived in hypoxic to anoxic shelf environments. Thus they conclude that the shift of *Vampyromorpha* from shallow to deeper water was already effective during the Oligocene. The evolution of cephalopods remains rather enigmatic and the increase in work on fossils in this field over the last fifteen years or so is very stimulating for neontological studies. The article provides a case of migration from shallow to deep environments considered as refuges during periods of environmental crisis. So it may be of interest to a large community of evolutionary biologists.

The paper is very interesting and I just have two main points that need to be clarified in order to be fully convinced by the conclusions.

1a: First, the reasoning of the paper is based on the phylogenetic position of the species under study. The reader needs to be better convinced by this position.

In other words, why is this gladius really close to *Vampyroteuthis* if one does not take into account its stratigraphic position? Why can't it belong to an extinct *Loligosepiina* lineage? Why can't it belong to a peculiar group of *Teudopseina* considering the lack of knowledge in the late Mesozoic and Cenozoic coleoid fossil record?

Indeed, the description of gladius is very succinct and deserves to be better argued. The description given line 104 and 105 is weak to justify the affinity with *Loligosepiids*. This does not mean that it is incorrect, but this cannot be treated so succinctly since it is the basis of the paper. The diagnosis of *Loligosepiid* from the *Coleoid Treatise* (Fuchs 2016) or in Fuchs and Weis (2008) indicate: "Gladius with triangular median field and ventrally reduced (cup-shaped) conus. Hyperbolar zone length clearly exceed half gladius length". Or "Hyperbolar zones can be either as long as the lateral fields or distinctly longer, but the hyperbolar zone regularly exceeds at least 50% of the median field length (hyperbolar zone length/median field length >0.5)".

1b :Figure 1 and 2 are of very good quality. It's worth to fully exploit them in indicating precisely

the limits and extensions of the hyperbolar zones and the lateral fields, (for example underlined the exact position with dash line instead of hbz, mf lf and not only on the drawing Fig.1c). The legends are too vague to make the work reusable (use measurements for example or compare to other fossils). Profile images of the tomographic reconstruction could also help for the shape of the conus.

Explain why the hyperbolar zone length is closer to Vampyroteuthids than to teudopseina?

1c: The authors should further discuss comparisons with teudopseina (e.g. since they consider Necroteuthis to be a Loligosepiina with a lateral field and hyperbolar zones abnormally short, why can't Necroteuthis be a Teudopseina with no pointed median field and an enlarged anterior median field). Teudopseina have variable gladius (for example late Teudopseina Actinosepia (late Cretaceous) exhibits a large anterior median field). This point is briefly mentioned l166 but not really justified or discussed.

1d: Line 160, in the discussion, it is mentioned that Necroteuthis is morphologically intermediate between Mesozoic loligosepiids and extant Vampyroteuthis. In fact, it's hard to be convinced that it's intermediate, to me it appears different with a mix of Teudopseina and Loligosepiina characters. I don't understand what justifies "intermediate".

1e: In Sutton et al. 2015 phylogenetic hypothesis, Loligosepiids aren't the closest relative to Octopodiformes. This is neither mentioned nor discussed in the paper.

2: Second, the coleoids fossil record is rather incomplete and depends mainly on exceptional preservation deposits which are often hypoxic or anoxic environments. Do the authors consider that this may bias interpretations regarding migration to hypoxic environments? Providing a synthesis of the composition of the coleoid fauna by locality could help the discussion.

Some suggestions/questions or corrections:

3: SEM observations should be moved to SI as they are useful for the specialist but not to follow and assess chains of arguments developed in the paper.

4: Figure 4 : can moved to SI. Same reasons.

5: Some details in micropaleontological analysis should also be moved in SI as Shannon index as it is mentioned for foraminifera and not for nannoplakton.

6: L64 : for morphology-based phylogeny Lindgren et al. 2004 have to be cited.

7: L68 : it would be better here to talk about ghost taxa or ghost lineages than Lazarus "effect".

8: L 71 72: Sutton et al. 2015 or Kruta et al. 2016 analyses do not support this hypothesis. It is worth mentioning and discussing.

9: L 198 low BFOI in Tard Clay Fm and line 218 negative BFOI in the Tard Caly Fm, it is unclear.

10: L 230 : "As the Recent vampire squid habitat is exclusively stenohaline (as other coleoid cephalopods), Necroteuthis clearly did not migrate to the upper surface layer with very low salinity". Why this assertion?

10:

L 41 : correct "oceanic minimum zone" : oxygen

11: L 278 : Is Proteroctopus a Loligosepiid? It has been positioned as a basal Octopodiforme in Kruta et al. (2016).

12: L 305 : "There are no loligosepiid records from similar conditions represented by later OAE2 and OAE3 (Upper Cretaceous). This absence may indicate migration of vampyromorphs into the deep sea at the end of the Early Cretaceous." Considering the scarcity of fossils, isn't that a bit speculative?

13: L 333, 338 : the use of "exaptation" and "preadaptation" is confused. What are the arguments

in favour of exaptation, what exactly are we talking about (which characters) and why referring to the exaptation hypothesis?

Responses to reviewers:

We have followed all recommendations of all three reviewers. We would like to thank very much to all reviewers for constructive reviews, valuable remarks and comments. These significantly increased the quality of the text. All changes, additions and corrections are highlighted by blue below as well as in the MS. Newly added citations explaining and supporting in more details our investigations are highlighted by yellow in the text.

Sincerely,

Martin Košťák (corresponding author), on behalf of authors team

Dr. Martin Kostak, Ph.D., associate professor
Institute of Geology and Palaeontology
Faculty of Science, Charles University, Prague
Albertov 6, Prague 2, 128 43
Czech Republic
tel: +420 221951456, +420 776492395
e-mail: kostys@centrum.cz, martin.kostak@natur.cuni.cz

Reviewer #1

Line 43: accepted „likely“

Line 94: corrected „was“

Line 97: corrected „triggered“

Line 230: corrected „very likely“

Line 257: corrected „might coincide“

Line 354: corrected „dot“

Line 367: added „No. M59/4672 (Hungarian Natural History Museum)“

Line 394: added „(i.e. bathyal or deep-sea in biological terminology (Oji, 2001).“

Line 395-399: Micro CT: Address of storage space added into „**Supplementary materials**“: The Micro CT datasets generated during the current study are available in the web link to datasets at persistent repository housed the Faculty of Science, Charles University, Prague:

<https://drive.google.com/drive/folders/1U9kjudlIKxbFsvDW3r2xukPSa-9afvpz?usp=sharing>

Points to answer:

Lines 42-44: Yes, we agree, however, we will not discuss this point in the abstract with references. This discussion is briefly and newly added into Chapter „Introduction“.

Line 49: Two hypothesis explaining preferential occurrences of coleoids are newly added in Chapter „Introduction“ – after line 69.

The successive shift of cephalopods into deeper part of oceans tends to be explained with hypotheses that postulate exclusion from shallower habitats driven by higher biotic or abiotic pressures (Packard, 1972; Nixon et al., 2003). The first hypothesis suggested that coleoids in shallower waters were effectively outcompeted and colonized deeper environments with

smaller biotic pressures (explaining survivor strategy in in nautiloids). Some lineages were subsequently able to reinvaded shallower waters (Packard, 1972). The second hypothesis suggested that although coleoids inhabited both shallow and deep habitats, extinctions preferentially occurred in shallower environments, and the temporal shift in the preference for deeper habitats is simply indirectly driven by higher extinction rate in shallower environments. Hereby, we suggest that active specialization to deep-sea habitats with anoxic conditions indicate that bathymetric variability in origination rate is also important in explaining the long-term trends in the bathymetric distribution of octobranchians.

Lines (55)58 and 64: Three references are added in the Line 55, after word anoxia – yes, new studies especially in frequently recorded belemnites show interesting responses to environmental stress (i.e. anoxia).

Line 68: Lazarus effect – right, some records in the past were confused with other Decabrachians as well as with other Octobranchians, but they have recently been clearly identified (numerous works of Fuchs, summarized recently in Treatise - Coleoidea). Recognizing and detecting the deep sea fossil ecosystems is difficult as also stated by Oji (2001) – we have revised this issue in the Chapter Discussion as follows:

„Deep-water sediments of these periods are considerably rarer, therefore, it is possible that the extent of the preserved deep-sea sediments with exceptional preservation is still insufficient to detect this group (Oji 2001).“

Lines 85-87: ... Added Fuchs (2020)

Line 147: Added additional background: „Sediment samples used for the stable isotope analysis ($\delta^{13}\text{C}$, $\delta^{18}\text{O}$) were drilled from a homogeneous, diagenetically unaltered component of sediment. Samples with bioclasts, signs of recrystallization, cements and carbonate veins were excluded.

Line 193: Added: „...based on detailed tectonical and sedimentological analyses“

Line 252: added sentence: „However, morphological novelties based on cryptic specializations may be generated under anoxic conditions in both nearshore and offshore habitats (Wood and Erwin, 2018).“

Line: 306-307: One sentence have been added informing about vampyro disappearing. Sampling bias as well as preservational potential are newly discussed in Chapter Discussion:

„The preservational potential of coleoid bodies in the fossil record is strongly limited by their fragile remains and by large amount of ammonia concentrated in their soft tissues, inhibiting precipitation of authigenic minerals, Clements et al., 2017). However, coleoid gladii that represent a taphonomic control for loligosepiids occur in the Lower Turonian shelf sediments (Bohemian Cretaceous Basin) deposited under well-oxygenated conditions (Košťák et al. 2020), indicating that loligosepiids may retreated from shallower environments already during the Cretaceous. The absence of loliginid gladii suggests that loligosepiids may retreated from shallower environments already during the Cretaceous. Shallow- water sediments representing later multiple ocean anoxic events (OAE2, OAE3) did not provide any loligosepiids records yet. Deep-water sediments of these periods are considerably rarer, therefore, it is possible that the extent of the preserved deep-sea sediments with exceptional preservation is still insufficient to detect this group (Oji 2001).“

Line: 324: Reference of Tajika et al. (2018) has been added.

General points:

1) Establishment of deep-sea coleoids: traditionally, two hypotheses could explain the preferential occurrence of vampyromorphs in the deep sea (or coleoids versus nautiloids more generally) which are not mutually exclusive (Nixon et al., 2003; Packard, 1972) – shift in distribution from shallow water and/or occurrence in both deep and shallow water with extinction in shallow water. It is quite

interesting that what you are suggesting here is also to some degree reminiscent what happen with the nautiloid lineage which was initially more restricted to shallow water. It is hard to distinguish between these scenarios in the fossil record (Oji, 2001) but you provide quite good arguments for the former. I would like to see these two hypotheses/possibilities a bit more development in the introduction with historical references. In this context, also the latest research concerning the importance of shifts in redox conditions you are already alluding too could also be integrated (Wood and Erwin, 2018).

It is difficult to distinguish between these – but we think that these hypotheses need also to account for bathymetric variability in origination rate. We have revised the Introduction as follows:

„The successive shift of cephalopods into deeper part of oceans tends to be explained with hypotheses that postulate exclusion from shallower habitats driven by higher biotic or abiotic pressures (Packard, 1972; Nixon et al., 2003). The first hypothesis suggested that coleoids in shallower waters were effectively outcompeted and colonized deeper environments with smaller biotic pressures (explaining survivor strategy in in nautiloids). Some lineages were subsequently able to reinvaded shallower waters (Packard, 1972). The second hypothesis suggested that although coeloids inhabited both shallow and deep habitats, extinctions preferentially occurred in shallower environments, and the temporal shift in the preference for deeper habitats is simply indirectly driven by higher extinction rate in shallower environments. Hereby, we suggest that active specialization to deep-sea habitats with anoxic conditions indicate that bathymetric variability in origination rate is also important in explaining the long-term trends in the bathymetric distribution of loligosepiids.“

Wood and Erwin, 2018 – difficult to apply, we have not enough evidences.

2) Preservation and collection potential of vampyromorphs: I would be crucial to have a little bit more background on the preservation and collection potential of vampyromorphs versus Decabrachia and for coleoids more generally (Clements et al., 2017). A bit more reference that historically various shells have been wrongly attributed to Decabrachia would also be helpful to place your study into context. This is trivial for coleoid workers but would make even more relevant for researchers beyond this field.

A comprehensive summary of what we know about fossil gladii and their preservation:

Fuchs (2016): Treatise chapter 9B: The gladius and gladius vestiges in fossil Coleoidea. Donovan (2016): Treatise chapter 9C: Composition & Structure of Gladii in fossil Coleoidea

3) Please provide essential data from the tomography: The tomography of the specimen is crucial to reject its previous assignment to Sepiidae and to be able to scientifically reproduce your results.

Not only, also the SEM provided clear evidences it is not a cuttlebone...

It would be standard practice to at least provide the raw full resolution image stack and metadata and recommended to also provide the prepared dataset (Davies et al., 2017). Platforms like Zenodo and other more specialized databases (e.g., morphosource.org) provide storage space and wide range of possibilities.

Address of storage space: The Micro CT datasets generated during the current study are available in the persistent web link to datasets housed at the Faculty of Science, Charles University, Prague: <https://drive.google.com/drive/folders/1U9kjudlKxbFsvDW3r2xukPSa-9afvpz?usp=sharing>

4) Additional information on the degree of diagenesis and preservation (e.g., cements) of isotope samples: it would be crucial to have some additional background information on the preservation and diagenesis (Cathodoluminescence) of the samples using for isotope analysis.

We have added in the Methods that sediment portions affected by veins or containing any cements were carefully removed. In the Results, we have added that:

„The bulk carbonate samples show no significant correlation between $\delta^{13}\text{C}$ and $\delta^{18}\text{O}$ ($r \sim 0.33$) and are isotopically close to bulk samples from the Tard Clay Formation analyzed by Bechtel et al. (2012).”

Additionally the bulk carbonate samples show no significant correlation between $\delta^{13}\text{C}$ and $\delta^{18}\text{O}$ ($r \sim 0.33$; $r^2 \sim 0.1113$). We also note that the values of the $\delta^{13}\text{C}$ and $\delta^{18}\text{O}$ are fully in accordance to bulk-sample values published in great details by Bechtel et al. (2012) from identical interval and sediments of the Tard Clay Formation.

5) Additional references: some additional statements are made (concerning indicators of little transport of the fossil) and definition used (e.g., Lazarus effect) which needs to be backed up with references.

We have added this clarification into Introduction:

„The Lazarus effect (Fara, 2001) can either reflect a decline in the outcrop area of post-Cretaceous deep-sea oxygen-depleted habitats and/or a decline in geographic range or in total population size of vampire squids so that their preservation potential is reduced even when the outcrop area remains the same.“

Ref: Fara, E., 2001. What are Lazarus taxa?. *Geological Journal*, 36(3-4), pp.291-303.

6) Please state briefly the main arguments for this depth (organisms, sedimentary properties, etc.) - this would be useful to the readers.

The paragraph was rewritten and significantly improved according to reviewer comments. The arguments for the paleodepth reconstruction were published several times by various authors (e.g. Charbonnier S., 2009: le Lagerstätte de La Voulte, un environnement bathyal au Jurassique, *Mémoires du Muséum national d'Histoire naturelle*, 199, 272 p.). We have revised this as follows: „The bottom-water conditions at the site of the soft-bottom deposition, with soft-tissue mineralization in the sediment zone with sulfate reduction (Wilby et al. 1996), were probably oxygen-depleted and temporarily anoxic, with limited bioturbation and mass mortalities documented by pavements of epibenthic bivalves (*Bositra*) (Etter 2002). Charbonnier et al. (2007) showed that photophilic encrusters are missing, invertebrates are encrusted by non-photozoans groups such as serpulids, cyrtocrinid crinoids, sponges, and thecideid brachiopods, and the actualistic distribution of sea-spiders, some crustaceans and sea-stars indicate that the sedimentation took place on the outer shelf at ~200 m with dysphotic or aphotic conditions (Charbonnier et al., 2007).”

Potentially, but in the paper itself they are not very certain either way. Also they state some environments could be more illuminated and does not necessarily mean all fossils derive from such an environment. Please express yourself more carefully.

This is already obvious when just writing the abstract - which speaks about an autochthonous origin or a combination of multiple environments (there preferred hypothesis).

"The eyes, mostly covered in hexagonal facets are interpreted as either apposition eyes (poorly adapted to low-light environment) or, less likely, as refractive or parabolic superposition eyes (compatible with dysphotic palaeoenvironments). The interpretation that *V. parvulus* had apposition eyes suggests an allochthonous, shallow water origin. However, the presence of thecideoid brachiopod ectosymbionts on its carapace, usually associated to dim-light palaeoenvironments and/or rock crevices, suggests that *V. parvulus* lived in a dim-light setting. This would support the less parsimonious interpretation that *V. parvulus* had superposition eyes. If we accept the hypothesis that *V. parvulus* had apposition eyes, since the La Voulte palaeoenvironment is considered deep water and had a soft substrate, *V. parvulus* could have moved into the La Voulte Lagerstätte setting. If this is the case, La Voulte biota would record a combination of multiple palaeoenvironments."

Also this comment was accepted, the paragraph continues as follows: „However, the co-occurrence of crustaceans with eyes adapted to photic conditions (Vannier et al., 2016) with groups with dysphotic or aphotic preferences indicate that the total assemblage represents a mixture of bathymetrically-

distinct habitats (Audo et al. 2019), although any postmortem transport had to be minor and rapid as indicated by the excellent preservation of complete skeletons of fragile organisms. Fault-controlled escarpments with sponge communities not far from the site with the exceptional preservation (Charbonnier et al., 2007) indicate steep topographic gradients over short distances, and it is likely that even a limited short-distance migration to soft-bottom habitats in the wake of anoxic events or postmortem transport can explain the mixture of groups differing in ecological requirements in deeper environments. To summarize, coleoids in these environments still inhabited outer shelf close to the shelf/slope margin during the Callovian, and did not yet expand to deeper bathyal environments.“

Reviewer #2

Line 26: „living fossil“ deleted – the abstract has been shortened to 150 words (from 225)

Line 28: improved

Line 33: corrected „vampyromorph gladius“

Lines 50-54: The long sentence has been divided.

Line 76: added ... described by Kretzoi²⁰ as *Necroteuthis hungarica*

Line 77: added „and fills the gap in the fossil record“

Lines 90-92: improved

Line 106: „the Recent“ removed

Line 182: removed text in the brackets

Line 237: added word „later“

Line 249: We have added this clarification:

„On the one hand, although the roots of some present-day deep-sea invertebrate lineages, such as ophiomycetid ophiuroids or pterasterid and benthoplectinid asteroids, can be traced back to the Jurassic (Thuy et al. 2014)“

Ref: Thuy, B., Kiel, S., Dulai, A., Gale, A.S., Kroh, A., Lord, A.R., Numberger-Thuy, L.D., Stöhr, S. and Wisshak, M., 2014. First glimpse into Lower Jurassic deep-sea biodiversity: in situ diversification and resilience against extinction. *Proceedings of the Royal Society B: Biological Sciences*, 281(1786), p.20132624.

Line 263: The sentence has been added

„We propose the following geological timeline by documenting the occurrences of vampyromorphs in multiple Mesozoic Lagerstätten to explore the potential factors that may have driven vampyromorphs to adapt toward oxygen-depleted conditions in deep-sea environments.“

Line 263: „smaller“ changed to „less“

Line 347: added „extant“

Line 350: left „at least during“ – as this shift should happen earlier.

Reviewer #3

Line 41: corrected to „oxygen“ minimum zone

Line 64: Lingren et al 2004, has been added

Line 68: It would be better here to talk about ghost taxa or ghost lineages than Lazarus "effect". Lazarus taxon simply refers to a long gap in the fossil record of a given species/lineage. Ghost lineages are interpreted on the basis of missing ancestors in phylogeny. Above the species level, one can say that the Lazarus taxon is also a ghost lineage. But we keep just to the Lazarus taxon.

We have explained this as: „The Lazarus effect (Fara, 2001) can either reflect a decline in the outcrop area of post-Cretaceous deep-sea oxygen-depleted habitats and/or a decline in geographic range or in total population size of vampire squids so that their preservation potential is reduced even when the outcrop area remains the same.“

Lines 71-72: Sutton et al. 2015 or Kruta et al. 2016 analyses do not support this hypothesis. It is worth mentioning and discussing.

Sutton et al. 2015 - Sutton is not a coleoid expert (the expertise does not really matter, stick with actual arguments – see below). His study is based on a questionable character acquisition such as gladius measurements with erroneous anterior/posterior orientations. Therefore, this study is not generally accepted in coleoid researchers community.

Lines 71-72: Sutton et al. 2015 or Kruta et al. 2016 analyses do not support this hypothesis. It is worth mentioning and discussing.

Kruta et al. 2016 - Kruta et al. confirmed earlier assumptions whereupon *Proteroctopus* from the Callovian deep sea deposits of La Voulte possessed a well-developed gladius. The poorly preserved gladius is however not determinable; its systematic affinities are questionable. We assume, this may belong to loligosepiids and it is involved into this group in our MS. Apart from this, we don't exclude the option that deep sea migrations occurred several times before the Vampyroteuthina. Reference to Kruta et al. 2016 has been added into .

Line 198: low BFOI vs line 218 negative BFOI.

We explained in the chapter „Setting and methods“, last paragraph Microfossil investigation that the BFOI can vary between -100 to +100. Low BFOI (-48) means also slightly lower than interval -40 to 0, indicating dysoxic conditions.

Line 230: “As the Recent vampire squid habitat is exclusively stenohaline (as other coleoid cephalopods), *Nectoteuthis* clearly did not migrate to the upper surface layer with very low salinity”. Why this assertion?

changed to „very likely“

Line 278: Is *Proteroctopus* a Loligosepiid? It has been positioned as a basal Octopodiforme in Kruta et al. (2016).

Indeed, Kruta et al. (2016) found *Proteroctopus* to be a basal octobranchian (=vampyropod). We personally assume (as far as can be judged) loligosepiid affinities, but what does a Callovian ?deep sea migration tells us about? Later loligosepiids were again shallow water inhabitants and we clearly state that octobranchians were preadapted to hypoxia.

Line 305: “There are no loligosepiid records from similar conditions represented by later OAE2 and OAE3 (Upper Cretaceous). This absence may indicate migration of vampyromorphs into the deep sea at the end of the Early Cretaceous.” Considering the scarcity of fossils, isn't that a bit speculative?

Yes, we agree but we say also B: they are not present also in both anoxic and non-anoxic shelf lagerstätten/sediments, where other coleoids are present. Discussion is extended, some new arguments and references of Clements et al., 2017; Košťák et al., 2020 and Oji, 2011 have been added.

Lines 333 and 338: the use of “exaptation” and “preadaptation” is confused. What are the arguments in favour of exaptation, what exactly are we talking about (which characters) and why referring to the exaptation hypothesis?

We avoid these terms to avoid confusion (as there is no need to specify them for the sake of our analyses) – we have simply replaced them with „adaptation“.

General points

1a: First, the reasoning of the paper is based on the phylogenetic position of the species under study. The reader needs to be better convinced by this position.

In other words, why is this gladius really close to *Vampyroteuthis* if one does not take into account its stratigraphic position? Why can't it belong to a n extinct Loliasepiina lineage? Why it can't belong to a peculiar group of Teudopseina considering the lack of knowledge in the late Mesozoic and Cenozoic coleoid fossil record?

Answers and more details: The Octobranchia Treatise will soon be online & the diagnosis has been adjusted. Fuchs, D. 2020. Part M, Coleoidea, Chapter 23G: Systematic Descriptions: Octobranchia. Treatise Online 138:1-52. Added to references.

Line 85: Indeed, the description of gladius is very succinct and deserves to be better argued. The description given line 104 and 105 is weak to justify the affinity with Loliasepiids. This does not mean that it is incorrect, but this cannot be treated so succinctly since it is the basis of the paper.

Owing to a triangular median field flanked by well-developed hyperbolar zones, *Necroteuthis* is most likely affiliated to both extinct loliasepiids and extant *Vampyroteuthis*. Most important mutualities are mentioned & there is no reason to assume teudopseid affinities.

The diagnosis of Loliasepiid from the Coleoid Treatise (Fuchs 2016) or in Fuchs and Weis (2008) indicate: “Gladius with triangular median field and ventrally reduced (cup-shaped) conus. Hyperbolar zone length clearly exceed half gladius length”. Or “Hyperbolar zones can be either as long as the lateral fields or distinctly longer, but the hyperbolar zone regularly exceeds at least 50% of the median field length (hyperbolar zone length/median field length >0.5)”.

Triangular median field flanked by well-developed hyperbolar zones is one of the most important morphological features, We assume the editor is happy with the extent we discussed this issue. More on morpho-phylo discussion will be boring for the reader – and it is referenced (with some new issues). The Octobranchia Treatise will soon be online & the diagnosis has been adjusted. Fuchs, D. 2020. Part M, Coleoidea, Chapter 23G: Systematic Descriptions: Octobranchia. Treatise Online 138:1-52.

1b :Figure 1 and 2 are of very good quality. It's worth to fully exploit them in indicating precisely the limits and extensions of the hyperbolar zones and the lateral fields, (for example underlined the exact position with dash line instead of hbz, mf lf and not on only on the drawing Fig.1c).

Changed, dashed lines are added, length of hyperbolar zones is newly indicated, scale bar within the reconstruction of the gladius is presented.

Fig. explanation: Gladius of *Necroteuthis hungarica* Kretzoi, 1942 (holotype - specimen No. M59/4672 Hungarian Natural History Museum). **a** Nearly complete gladius, dorsal view, scale bar = 2 cm. **b** Detail of the apical part forming conus (c), mf – median field, lf – lateral fields, hbz – hyperbolar zones, dashed lines mark the hyperbolar zones separating lateral fields from the median field, scale bar = 2 cm. **c** Reconstruction of the gladius, red lines demarcate hyperbolar zones, mf – enlarged median field, lhf – length of hyperbolar zones, scale bar = 10 cm, rectangle shows the position of the figure **1d**, scale bar = 2 cm. **d** Detail of the lateral field with hyperbolar zone and median field, scale bar = 2 cm. **e** Detail of the median field with marked concentric growth lines (gl).

The legends are too vague to make the work reusable (use measurements for example or compare to other fossils). Profile images of the tomographic reconstruction could also help for the shape of the conus.

The gladius is compacted/compressed. This is mentioned in the text. – therefore the deformation of the conus is also seen in the micro CT visualisation. The legends are more properly explained and extended.

Explain why the hyperbolar zone length is closer to Vampyroteuthids than to teudopseina?

The hyperbolar zone of *Necroteuthis* is closer to Vampyroteuthis than to Teudopseina, because *Necroteuthis* and *Vampyroteuthis* are phylogenetically closer (see also point 1d).

1c: The authors should further discuss comparisons with teudopseina (e.g. since they consider *Necroteuthis* to be a Lologosepiina with a lateral field and hyperbolar zones abnormally short, why can't *Necroteuthis* be a Teudopseina with no pointed median field and an enlarged anterior median field). Teudopseina have variable gladius (for example late Teudopseina *Actinosepia* (late Cretaceous) exhibits a large anterior median field).

Actinosepia is an extremely specialized Late Cretaceous coleoid with specific habitat linked to the earliest sea-grasses (see also Forsey, 2020). The gladius resembles also sepiid cuttlebone, which is probably connected to similar habitat – i.e. high level of convergency.

This point is briefly mentioned 1166 but not really justified or discussed.

Teudopseina have a pointed gladius and are therefore not worth to be extensively considered. Additionally, during the phylogeny of teudopseids, the median field is rather reduced than enlarged. Detailed comprehensive morphological and phylogenetical analyses are beyond the scope of the present contribution.

1d: Line 160, in the discussion, it is mentioned that *Necroteuthis* is morphologically intermediate between Mesozoic lologosepiids and extant Vampyroteuthis. In fact, it's hard to be convinced that it's intermediate, to me it appears different with a mix of Teudopseina and Lologosepiina characters. I don't understand what justifies "intermediate".

The referee is focused on Teudopseina, but this group is fundamentally different from *Necroteuthis*. The referee refers to the possibility that a second group of octobranchians with a fully developed gladius (besides Vampyromorpha) survived the K-Pg boundary. Such a scenario has no support. Teudopseina is characterized by the reduction of the median field rather than enlargement. More additional informations are given in Fuchs (2020), a newly referenced publication in the text.

1e: In Sutton et al. 2015 phylogenetic hypothesis, Lologosepiids aren't the closest relative to Octopodiformes. This is neither mentioned nor discussed in the paper.

The study of Sutton is based on a questionable character acquisition such as gladius measurements with erroneous anterior/posterior orientations. Therefore, we think his hypothesis is not reliable and as we have the limited word count, we avoid discussing it. We think that comprehensive cladistic analyses is beyond the scope and subject to an upcoming paper.

2: Second, the coleoids fossil record is rather incomplete and depends mainly on exceptional preservation deposits which are often hypoxic or anoxic environments. Do the authors consider that this may bias interpretations regarding migration to hypoxic environments?

Sentences and enlarged discussion are added into Chapters: Introduction and Discussion:

„The preservational potential of coleoid bodies in the fossil record is strongly limited by their fragile remains and by large amount of ammonia concentrated in their soft tissues, inhibiting precipitation of authigenic minerals, Clements et al., 2017). However, coleoid gladii that represent a taphonomic control for lologosepiids occur in the Lower Turonian shelf sediments (Bohemian Cretaceous Basin)

deposited under well-oxygenated conditions (Košťák et al. 2020), indicating that loligosepiids may retreated from shallower environments already during the Cretaceous. The absence of loliginid gladii suggests that loligosepiids may retreated from shallower environments already during the Cretaceous. Shallow- water sediments representing later multiple ocean anoxic events (OAE2, OAE3) did not provide any loligosepiids records yet. Deep-water sediments of these periods are considerably rarer, therefore, it is possible that the extent of the preserved deep-sea sediments with exceptional preservation is still insufficient to detect this group (Oji 2001).“

Providing a synthesis of the composition of the coleoid fauna by locality could help the discussion.

This is an interesting idea for the further research – it would be interesting to know ratios between teudoseids, loligosepiids and other coleoids. However, such a synthesis explodes the main scope of the present contribution.

3: SEM observations should be moved to SI as they are useful for the specialist but not to follow and assess chains of arguments developed in the paper.

4: Figure 4 : can moved to SI. Same reasons.

5: Some details in micropaleontological analysis should also be moved in SI as Shannon index as it is mentioned for foraminifera and not for nannoplakton.

We wish to keep SEM, figure 4 and Shannon index (within micropaleontological part) in the main text for the following reasons: The SEM is a crucial method showing clear difference from sepiid cuttlebone and subsequently indicating a gladius character and state of preservation. The Fig. 4 – nannoplankton accumulations are a very nice examples of the planktic blooms we are talking about in the text. Moreover, framboidal pyrite indicates bacterial activity in dysoxic conditions – another important point in our MS. Yes, Shannon index is mentioned for forams in one sentence, according to our specialists in micropalaeontology it is very important to keep it within the chapter. Micropalaeontological analysis. We assume, our paper will attract also interest of micropalaeontological specialists, as important results are also based on these analysis.

REVIEWERS' COMMENTS:

Reviewer #1 (Remarks to the Author):

Thank you for addressing my previous suggestions. I feel the manuscript is even easier to follow and of broader relevance. I only have some very minor additional things (mostly typos or the use of particular words; 2 additional reference which further back up some of the taphonomic scenarios you are suggesting) i noticed in the revised manuscript which might need to be addressed. All my points can be found as comments in the annotated.

It seems i forgot to sign my previous review by mistake.

Looking forward to seeing this holistic paleontological/paleobiological manuscript published.

Kenneth De Baets

Reviewer #3 (Remarks to the Author):

would like to thank the authors for adding additional information and clarifications. In the review I emphasized the need of clear and detailed description and comparison with other taxa because even if it is not the most exciting part of the paper, it remains essential data to share.

I still don't see what is in between or internediary vampyro-loligo in *Necroteuthis galduis* shape. Besides, in the treatise on can read: *Nectoteuthis* exhibits a mosaic of loligosepiid and vampyroteuthid characters.

I do agree that "mosaic" seems to be more obvious than the intermediary.

This is actually what is mentioned in the reference added (the new Treatise) : *Necroteuthis* from the Oligocene of Hungary supports this view as its gladius exhibits a mosaic of loligosepiid and vampyroteuthid characters. *Necroteuthis* is accordingly seen as the connecting link between the two vampyromorph suborders.

Does this mosaic of characters demonstrate a closest phylogenetic relationship to vampyroteuthis than other loligosepiid?

It would be a more cautionary position not to exclude the option that *Necroteuthis* could also belong to an extinct lineage of loligosepiid. The reasonable argument is that chances that several lineages survived to the K/T boundary are low, but I do not know if this can be ruled out considering the scarcity of the fossil record.

But anyway, the hypothesis of the article is attractive and it will be exciting to see if other data confirm it in the future.

Responses to Reviewer 3.

We have more emphasized that *Necroteuthis* is placed within the Vampyromorphina (rather than Loligosepiina), because of its deep sea life style too. Gladius "similarities" are phylogenetically ambiguous and thus secondary in this context.

- *"I still don't see what is in between or intermediat vampyro-loligo in Necroteuthis galduis shape. Besides, in the treatise on can read: Nectoteuthis exhibits a mosaic of loligosepiid and vampyroteuthid characters.*

I do agree that "mosaic" seems to be more obvious than the intermediary. This is actually what is mentionned in the reference added (the new Treatise) : Necroteuthis from the Oligocene of Hungary supports this view as its gladius exhibits a mosaic of loligosepiid and vampyroteuthid characters. Necroteuthis is accordingly seen as the connecting link between the two vampyromorph suborders. "

The referee accepts the statement made by Fuchs (2020; Treatise Online), in which *Necroteuthis* is treated as a connecting link between the Loligosepiina and Vampyromorphina. The referee apparently accepts the terms "mosaic" (of characters) and "connecting link", but rejects the term "intermediate". Though we consider "intermediate" as a common & neutral term in morphological descriptions/comparisons/discussions, we adopt "mosaic".

- *"Does this mosaic of characters demonstrate a closest phylogenetic relationship to vampyroteuthis than other loligosepiid?"*

We have rewritten the corresponding part. We now extensively discuss the differences of *Necroteuthis* & *Vampyroteuthis* on the one hand and Loligosepiina on the other. In this context, it is important to repeat that our classification is based on morphological, stratigraphical, and ecological implications. Phylogenetic/cladistic implications are (owing to still insufficient data) beyond the scope of the present work.

- *"It would be a more cautionary position not to exclude the option that *Necroteuhtis* could also belong to an extinct lineage of loligosepiid. The reasonable argument is that chances that serveral lineages survived to the K/T boudary are low, but I do not know if this can be ruled out considering the scarcity of the fossile record."*

We do not exactly understand what the referee target on by this critique point? To our best knowledge and belief, we cannot find convincing arguments for placing *Necroteuthis* within the Loligosepiina (however, some morphological features are obvious). By contrast, we consider such a classification (implying two independent deep sea migrations plus two independent K/Pg Crisis survivors) appears to us hard to be communicated.

Again, it is its deep sea life style (rather than distinct gladius characteristics) that primarily led us to classify *Necroteuthis* outside shallow water loligosepiids and instead in the Vampyromorphina. Apart from this, our systematic assignment does not affect our conclusion whereupon the entire vampyromorph lineage somewhen migrated to deeper waters.

M. Košťák and D. Fuchs (on behalf of author's team)

Prague, December 7, 2020